# Specification of female germline by microRNA orchestrated auxin signaling in *Arabidopsis*

Jian Huang[1,4], Lei Zhao[1,5], Shikha Malik[1,6], Benjamin R. Gentile[1], Va Xiong[1], Tzahi Arazi[2], Heather A. Owen[1], Jiří Friml [3] & Dazhong Zhao [1] ✉

Germline determination is essential for species survival and evolution in multicellular organisms. In most flowering plants, formation of the female germline is initiated with specification of one megaspore mother cell (MMC) in each ovule; however, the molecular mechanism underlying this key event remains unclear. Here we report that spatially restricted auxin signaling promotes MMC fate in *Arabidopsis*. Our results show that the microRNA160 (miR160) targeted gene *ARF17* (*AUXIN RESPONSE FACTOR17*) is required for promoting MMC specification by genetically interacting with the *SPL/NZZ* (*SPOROCYTELESS/NOZZLE*) gene. Alterations of auxin signaling cause formation of supernumerary MMCs in an *ARF17*- and *SPL/NZZ*-dependent manner. Furthermore, miR160 and *ARF17* are indispensable for attaining a normal auxin maximum at the ovule apex via modulating the expression domain of PIN1 (PIN-FORMED1) auxin transporter. Our findings elucidate the mechanism by which auxin signaling promotes the acquisition of female germline cell fate in plants.

A longstanding question in both plants and animals is how germline cells are specified to permit sexual reproduction. During the life cycle of flowering plants, the alternation between the diploid sporophyte and haploid gametophyte generations is connected by the specification of a germline post-embryonically in reproductive organs of the flower[1]. A single megaspore mother cell (MMC), designated as the first committed cell of the female germline lineage, normally originates from a somatic cell in one ovule, a part of the makeup of the female reproductive organ[2–4]. The MMC undergoes meiosis to yield four haploid megaspores, but only one megaspore named the functional megaspore (FM) survives and gives rise to the female gametophyte (called embryo sac or megagametophyte), which contains one haploid egg cell. Following double fertilization, the seed which typically harbors a single sexually produced embryo is eventually formed. Previous studies have identified several negatively acting regulatory pathways

that restrict the female germline to a single MMC per ovule. In *Arabidopsis*, the inactivation of *WUS* (*WUSCHEL*) gene by the RBR1 (RETINOBLASTOMA RELATED1) transcriptional repressor is essential for the MMC formation[5,6]. A class of cyclin-dependent kinase (CDK) inhibitors called KRPs (KIP-RELATED PROTEINs, also known as ICKs: INTERACTOR/INHIBITOR OF CYCLIN-DEPENDENT KINASEs) represses the inactivation of RBR1 via inhibiting CDKA;1. Functional disruption of *KRP/ICK* and *RBR1* genes causes additional mitotic divisions of differentiated MMC or its precursor, and thus the formation of supernumerary MMCs. In rice and maize, a Leucine-Rich Repeat Receptor-Like Kinase (LRR-RLK)-linked signaling pathway, including the receptor MSP1 (MULTIPLE SPOROCYTE 1) and its putative ligand TDL1A/MAC1 (TAPETUM DETERMINANT-LIKE 1A/MULTIPLE ARCHESPORIAL CELLS 1), prevents differentiation of somatic cells surrounding the MMC into multiple MMCs[7–9]. Similarly, in *Arabidopsis*, the

[1]Department of Biological Sciences, University of Wisconsin-Milwaukee, Milwaukee, WI 53211, USA. [2]Institute of Plant Sciences, ARO, Volcani Center, HaMaccabbim Road 68, 7505101 Rishon LeZion, Israel. [3]Institute of Science and Technology (IST) Austria, 3400 Klosterneuburg, Austria. [4]Present address: T3 Bioscience LLC, Mequon, WI 53092, USA. [5]Present address: Beijing Genesee Biotech, Inc, 101400 Beijing, China. [6]Present address: Department of Plant Pathology and Microbiology, Iowa State University, Ames, IA 50011, USA. ✉e-mail: dzhao@uwm.edu

cytochrome P450 protein KLU and the chromatin remodeling complex subunit SWR1 co-activate the expression of *WRKY28* in somatic cells surrounding the MMC, which suppresses them to acquire the MMC identity[10]. In addition, trans-acting small interfering RNAs known as tasiR-ARFs repress the expression of *ARF3* in cells neighboring the MMC to inhibit the formation of ectopic MMCs[11,12]. Moreover, the epigenetic regulation associated with *AGO9* (*ARGONAUTE9*), *RDR6* (*RNA-DEPENDENT RNA POLYMERASE 6*), DRM (DOMAINS REAR-RANGED METHYLASE), RNA helicase gene *MEM* (*MNEME*), and *SPL/NZZ* (*SPOROCYTELESS/NOZZLE*), might use the similar mechanism to control the number of MMC[13–22]. Although significant progress has been made toward understanding pathways that restrict MMC formation, the molecular mechanism underlying the promotion of MMC identity remains elusive.

The phytohormone auxin acts as a major regulator of patterning and adaptive development in plants[23]. Its unique property among signaling molecules is the formation of local maxima or gradients as a result of local biosynthesis and, in particular, the polar auxin transport (PAT) mediated by PIN (PIN-FORMED) auxin exporters[24–26]. Auxin accumulation in individual cells leads to developmental reprogramming via regulating expression of auxin-responsive genes by ARF (AUXIN RESPONSE FACTOR) transcription factors[27–30]. For example, during root development, xylem and phloem cells are derived from a single, bifacial stem cell. A local auxin-signaling maximum specifies the stem cell organizer by activating expression of *HD-ZIP III* genes (*CLASS III HOMEODOMAIN-LEUCINE ZIPPER*) via ARF5 [also known as MONO-PTEROS (MP)], ARF7, and ARF19[31]. So far, it is not known whether the positional information provided by auxin signaling is involved in MMC specification.

We previously showed that the *Arabidopsis MIR160a* (*MICRO-RNA160a*) gene loss-of-function mutant *foc* (*floral organs in carpels*) is defective in embryogenesis[32]. miR160 negatively regulates expression of *ARF10*, *ARF16*, and *ARF17*[33–35]. In *Arabidopsis*, the *spl/nzz* mutant fails to develop MMC[14,15,17]. Here we report that the miR160-targeted *ARF17* specifies the MMC by genetically interacting with *SPL/NZZ*. Auxin signaling is required for MMC formation in an *ARF17*- and *SPL/NZZ*-dependent manner, while miR160 and ARF17 define the expression domain of PIN1, which contributes to establishment of the local auxin maximum at the ovule apex. Our findings highlight the importance of the miRNA fine-tuned auxin signaling that controls specification of the initial female germline cell MMC in flowering plants.

## Results

### Specification of MMC by miR160 and *ARF17*
In this study, we found that development of 66.1% of ovules was arrested in the *foc* mutant (Fig. 1a, b and Supplementary Table 1). To investigate the cause of *foc* ovule abortion, we analyzed MMC differentiation morphologically using differential interference contrast (DIC) microscopy combined with examining expression of the MMC marker *pKNU::KNU-VENUS* by confocal microscopy[11,32,36]. We first studied MMC differentiation along with early ovule development in wild-type (WT) plants. The dome-shaped ovule primordium arises at stage 1-I (Supplementary Fig. 1a) and elongates at stage 1-II (Supplementary Fig. 1b). Weak to moderate *pKNU::KNU-VENUS* signals were detected in one subepidermal cell [named MMCP (MMC precursor)] at the distal end of ovule primordia at stages 1-I (Supplementary Fig. 1a) and 1-II (Supplementary Fig. 1b), respectively, although MMCPs are morphologically indistinguishable from other subepidermal cells. Ovules at stage 2-I lack integuments (Supplementary Fig. 1c), while the inner and outer integuments are initiated successively at stages 2-II (Supplementary Fig. 1d) and 2-III (Supplementary Fig. 1e). From stage 2-I to 2-III, MMCs are so designated because *pKNU::KNU-VENUS* signals are robust in individual MMCs which have larger cell and nucleus size than somatic cells surrounding them (Supplementary Fig. 1c–e). Additionally, we observed two MMC-like (MMCL) cells at stage 2-I at a very low

rate (Supplementary Fig. 1f, 4.8%, $n = 250$); however, *pKNU::KNU-VENUS* is predominantly expressed in only one MMCL cell. After stage 2-I, the *pKNU::KNU-VENUS* signal was detected only in one MMC. Our results suggest that MMC originates from MMCP and observations at pre-meiotic stages (stage 2-I to 2-III) are suitable for assessing MMC numbers.

Each WT ovule typically produces one MMC (Fig. 1e, q, 95.2%, $n = 250$, Fig. 1i, 94.4%, $n = 126$, and Supplementary Fig. 1c–e) at pre-meiotic stages; however, we observed two or more MMCLs in *foc* ovules (Fig. 1f, q, 25.7%, $n = 280$ and Fig. 1j, 24.6%, $n = 122$), suggesting that miR160 is important for MMC specification. miR160 negatively regulates expression of *ARF10*, *ARF16*, and *ARF17*[32–35]; therefore, to determine which *ARF* is involved in MMC specification, we generated *pARF10::mARF10*, *pARF16::mARF16*, and *pARF17::mARF17* transgenic plants in the L*er* background to express their miR160-resistant versions which are no longer subject to miR160's negative regulation under the control of their own promoters. Eighty percent of ovules and seeds were aborted in *pARF17::mARF17* plants (Fig. 1c and Supplementary Table 1). Notably, as opposed to normal ovules from *pARF10::mARF10* and *pARF16::mARF16* plants, only ovules from 12 examined *pARF17::mARF17* independent lines produced super-numerary MMCLs (Fig. 1g, q, 28.6%, $n = 287$ and Fig. 1k, 28.6%, $n = 140$), thus phenocopying *foc* ovules. When introduced into the *foc* background, *pARF17::mARF17 foc* plants showed more aborted ovules (Fig. 1d and Supplementary Table 1) and an even higher number of supernumerary MMCLs in all 15 examined independent plants (Fig. 1h, q, 56.0%, $n = 364$ and Fig. 1l, 56.6%, $n = 136$), possibly due to a dosage effect of *ARF17*. In addition, analysis of callose deposition that was used as a cytological marker for MMC undergoing meiosis[14,16] found that meiosis typically occurred only in one MMC in WT, *foc*, and *pARF17::mARF17* ovules (Fig. 1m, n, 92.6%, $n = 122$ and Fig. 1q), whereas two MMCs are preparing to enter meiosis in *pARF17::mARF17 foc* ovules (Fig. 1o, p, 18.9%, $n = 175$ and Fig. 1q). During later female gametophyte (FG) development[37], embryo sacs with various defects were observed both in *foc* and *pARF17::mARF17* plants (Supplementary Fig. 2a–s). Even two embryo sacs were observed in 5.6% of *pARF17::mARF17 foc* ovules (Supplementary Fig. 2t, $n = 216$). Taken together, our results suggest that the miR160-controlled *ARF17* is a key part of the machinery ensuring specification of a single MMC per ovule in *Arabidopsis*.

### Precise control of *ARF17* spatial expression is important for MMC specification
To understand how miR160 and its target *ARF17* control MMC specification, we first examined their expression during early ovule development. Whole-mount in situ hybridization studies showed that the *MIR160a* gene is primarily expressed in chalaza and funiculus of ovules at stage 2-III (Fig. 2a, e). *pMIR160a5'::NSL-3xGFP::MIR160a3'* transgenic plants also showed GFP signals mainly in chalaza at stage 2-II (Fig. 2g) and in both chalaza and funiculus at stage 2-III (Fig. 2h), confirming the in situ hybridization results. The mature miR160 was detected not only in chalaza and funiculus but also highly in MMC (Fig. 2b). To test where miR160 acts, we generated the miR160 GFP sensor[38] driven by the UBI10 promoter[39]. GFP signals were ubiquitously detected in *pUBI10::NSL-3xGFP* ovule cells, including the MMC (Fig. 2i, j, control). In the *pUBI10::miR160sensor-NSL-3xGFP* ovule, GFP signals were not observed in the MMC and became weaker in the chalaza (Fig. 2k, l), suggesting that the mature miR160 is active in a range of ovule cells, and in particular, the MMC.

*ARF17* transcripts were observed in the MMC, chalaza, and funiculus in the WT ovule (Fig. 2c, f), with an overall higher level in the *foc* ovule (Fig. 2d). The GFP signal was not observed in *pARF17::ARF17-GFP* ovules directly using a confocal microscope (Fig. 2m), however, the ARF17 protein was found in the MMC in a whole-mount immunofluorescence assay of *pARF17::ARF17-GFP* ovules (Fig. 2m, the bottom left inset), suggesting that a relatively low level of ARF17 protein in the MMC is

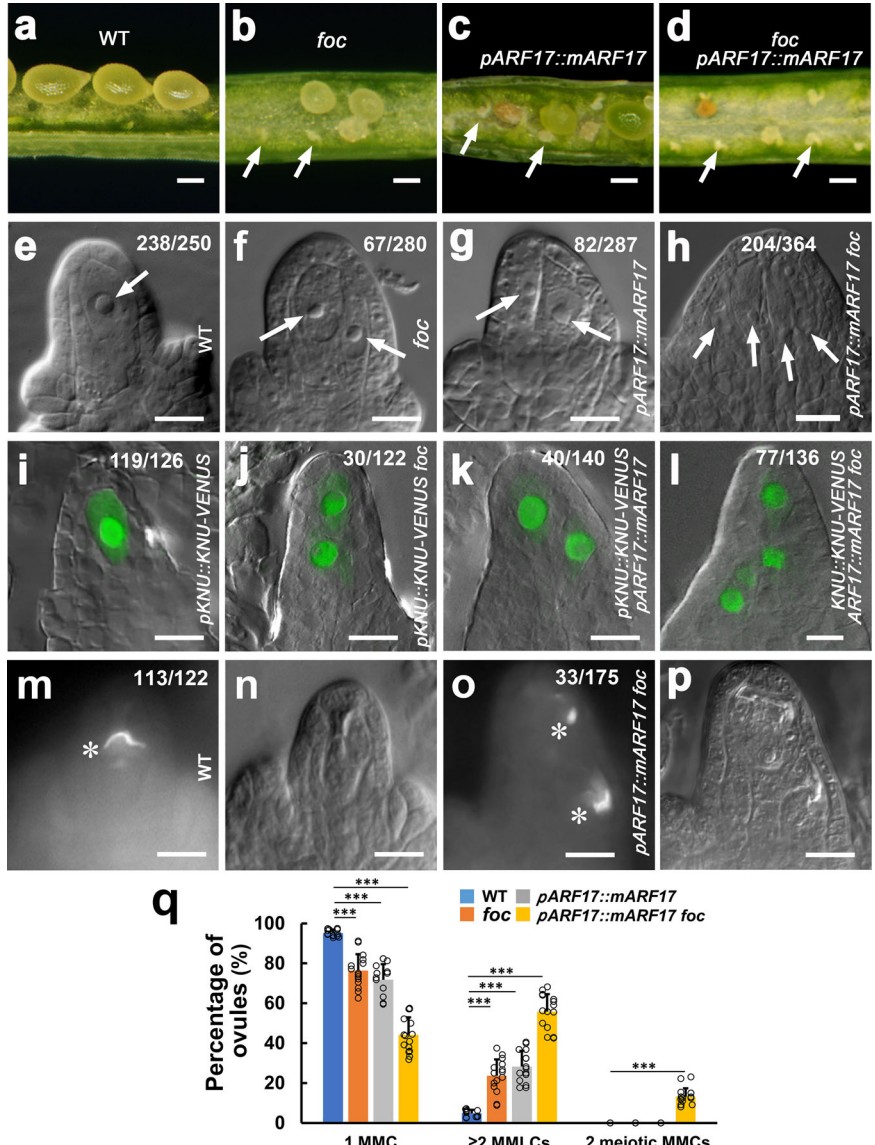

**Fig. 1 | miR160 and its target *ARF17* control MMC formation. a–d** Development of abnormal ovules. Developing seeds in the silique from the L*er* wild-type (WT) plant (**a**), and aborted ovules in siliques of *foc* (**b**), *ARF17::mARF17* (**c**), and *ARF17::mARF17 foc* (**d**) plants. Arrows indicate aborted ovules. **e–p** Formation of supernumerary MMCLs and MMCs. **e–h** Differential interference contrast (DIC) images of ovules showing MMCLs (arrows). **i–l** Merged confocal and DIC images of ovules expressing *pKNU::KNU-VENUS* marking the MMC fate. **m–p** Callose deposition indicating MMC undergoing meiosis. **e, i** A single MMC in the wild-type (WT) ovule. **f, j** Two MMCLs in the *foc* ovule. **g, k** Two MMCLs in the *pARF17::mARF17* ovule. **h, l** Four MMCLs in the *pARF17::mARF17 foc* ovule. **m–p** One MMC in the WT ovule (**m**) and two MMCs in the *pARF17::mARF17 foc* ovule (**o**) undergoing meiosis (denoted by white asterisks). **m, o** Callose staining. **n, p** DIC images of (**m, o**), respectively. **q** Quantifications of MMCs and MMCLs in WT, *foc*, *pARF17::mARF17*, and *pARF17::mARF17 foc* ovules from 15 individual plants (*n* = 15 plants; two-sided Student's *t* test; Error bars: SD; \**p* < 0.05, \*\**p* < 0.01, and \*\*\**p* < 0.001). Source data are provided as a Source data file. Numbers in the panels denote frequencies of phenotypes shown. Scale bars, 1.5 mm (**a–d**), 10 μm (**e–p**).

important for its specification. In contrast, under a confocal microscope ARF17 was readily detected in extra MMCLs of *pARF17::ARF17-GFP foc* (Fig. 2n), *pARF17::mARF17-GFP* (Fig. 2o), and *pARF17::mARF17-GFP foc* (Fig. 2p) ovules. Moreover, weak GFP signals were also present in chalaza and funiculus of these ovules. Our results suggest that the mature miR160 negatively regulates the expression of *ARF17*.

To test whether expression of *ARF17* and miR160 is required for ectopic MMCL formation, we first overexpressed miR160-resistant *ARF17* in the MMC using the MMCP and MMC specific promoter *pKNU*[41]. A majority of *pKNU::mARF17* ovules had one MMC, while two MMCs were observed in ~5% of *pKNU::mARF17* ovules (Fig. 2q, 5.2%, *n* = 212, Fig. 2r, 4.6%, *n* = 280, and Fig. 2u, v, 4.7%, *n* = 233). We then employed the STTM (Short Tandem Target Mimic) method[40] to knock down miR160 in the MMC using the *KNU* promoter. Similarly,

*pKNU::STTM160/160-48* ovules mainly produced one MMC, although 5% of ovules formed two MMCs (Fig. 2s, 4.9%, *n* = 306, Fig. 2t, 4.8%, *n* = 252, and Fig. 2w, x, 4.5%, *n* = 288). In WT, 4.8% of ovules also formed two MMCLs (Supplementary Fig. 1f), which is similar to *pKNU::mARF17* and *pKNU::STTM160/160-48* ovules, suggesting that miR160 restricts *ARF17* expression in ovule cells and overexpression of *ARF17* solely in the MMC does not promote MMC proliferation.

## *ARF17* is required for MMC specification by genetically interacting with *SPL/NZZ*

To test whether *ARF17* is required for MMC specification, we overexpressed *ARF17* in the *spl* mutant, which does not produce MMC[14,15,17]. Compared with the case in WT (Fig. 3a, 95.2%, *n* = 250 and Fig. 3e, i, 91.7%, *n* = 133), almost no MMC (Fig. 3b, 98.9%, *n* = 180 and Fig. 3f, j, 0%,

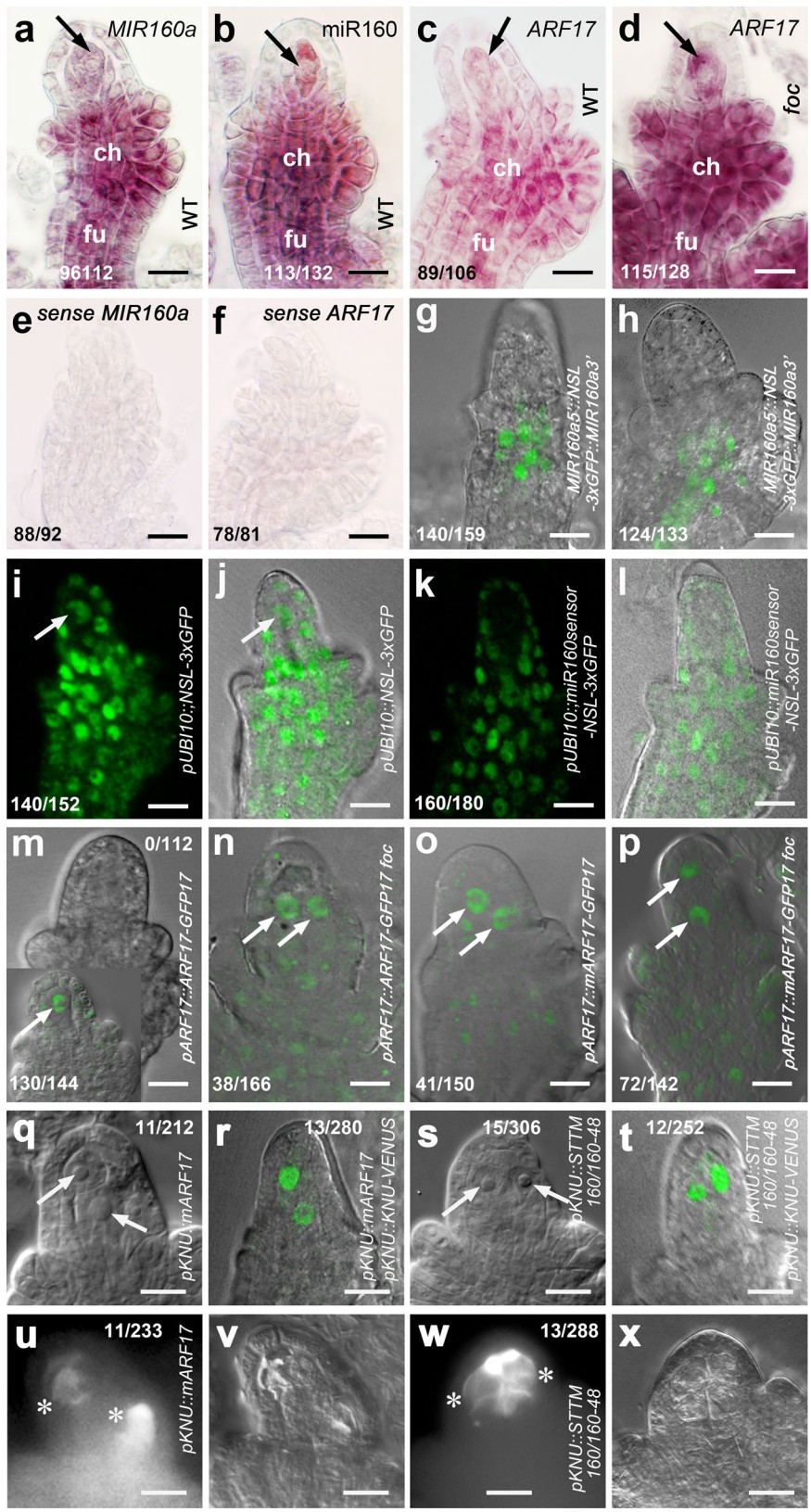

$n = 216$) was formed in *spl* ovules; however, *pARF17::mARF17* partially rescued the formation of MMC (Fig. 3c, 44.0%, $n = 241$ and Fig. 3g, k, 29.8%, $n = 141$) in the *spl* mutant. Similarly, the restored MMC differentiation was also observed in the *spl foc* double mutant (Fig. 3d, 35.2%, $n = 210$ and Fig. 3h, i, 30.9%, $n = 152$). No FM was formed in the *spl* mutant ovule (Fig. 3m, n); however, FM was found in

*pARF17::mARF17 spl* (Fig. 3o, 26.7%, $n = 277$) and *spl foc* (Fig. 3p, 24.4%, $n = 172$) ovules. Moreover, manual pollination led to seed development in both *pARF17::mARF17 spl* and *spl foc* plants in comparison with the *spl* mutant (Supplementary Fig. 3a–d), further suggesting that overexpression of *ARF17* could partially enable female gametogenesis in the *spl* mutant.

**Fig. 2 | Expression of miR160 and *ARF17* in the MMC is essential for its specification. a–f** Images of whole-mount mRNA in situ hybridization showing expression of the *MIR160a* (*FOC*) gene mainly in chalaza (ch) and funiculus (fu) (**a**) and the mature miR160 also in the MMC (**b**) in WT ovules at stage 2-III; *ARF17* in the MMC, ch, and fu in the WT ovule (**c**) but with the overall enhancement in the *foc* ovule (**d**) at stage 2-III. Arrows: MMC. Sense probes of *MIR160a* (**e**) and *ARF17* (**f**), respectively. Experiments were repeated twice with similar results. Merged confocal and DIC images of *pMIR160a5'::NSL-3xGFP::MIR160a3'* ovules showing GFP signals mainly in the chalaza at stage 2-II (**g**) and in both chalaza and funiculus at stage 2-III (**h**). Experiments were repeated three times with similar results. **i–l** Examination of the miR160 GFP sensor showing the mature miR160 acts in the MMC (arrows). Confocal (**i**) and merged confocal and DIC (**j**) images of the *pUBI10::NSL-3xGFP* ovule showing ubiquitous expression of GFP signals in ovule cells, including the MMC. Confocal (**k**) and merged confocal and DIC (**l**) images of the *pUBI10::miR160sensor-NSL-3xGFP* ovule showing no signal in the MMC and weaker signals in the chalaza. Experiments were repeated three times with similar results. **m–p** Merged confocal and DIC images showing localization of the ARF17 protein in MMCs/MMCLs (arrows). **m** Signal was not observed directly under the confocal in the *pARF17::ARF17-GFP* ovule at stage 2-III, while immunofluorescence assay showing signal in one MMC (the bottom left inset). Confocal microscope readily detected signals in two MMCs/MMCLs of *pARF17::ARF17-GFP foc* (**n**), *pARF17::mARF17-GFP* (**o**), and *pARF17::mARF17-GFP foc* (**p**) ovules at stage 2-III. Experiments were repeated three times with similar results. **q–x** Overexpression of *ARF17* in the MMC did not cause the MMC proliferation. **q, s** DIC images. **r, t** Merged confocal and DIC images of ovules expressing *pKNU::KNU-VENUS*. **q, r** Two MMCs (arrows) were produced at a low rate in the *pKNU::mARF17* ovule at 2-III. **s, t** Two MMCs (arrows) were formed at a low rate in the *pKNU::STTM160/160-48* ovule at stage 2-III. Callose deposition indicating two MMCs undergoing meiosis (denoted by white asterisks) in *pKNU::mARF17* (**u, v**) and *pKNU::STTM160/160-48* (**w, x**) ovules. **u, w** Callose staining. **v, x** DIC images of (**u, w**), respectively. Experiments were repeated three times with similar results. Scale bars, 10 μm.

To further test the requirement of *ARF17* for MMC specification, we generated three *crispr-arf17* (*carf17*) independent mutants (Supplementary Fig. 4a–c) and identified a weak *spl* T-DNA insertion allele *spl-3* (Supplementary Fig. 4j, k). The *carf17* mutants are normal in vegetative growth (Supplementary Fig. 4d, e), but male sterile (Supplementary Fig. 4f–i). The *carf17-6* mutant (still named as *carf17* in this paper for simplicity), which has no *Cas9*, was used for further analysis. Although the MMC differentiation is morphologically normal in the *carf17* mutant (Fig. 3q, 94.6%, *n* = 312 and Fig. 3u), failure of the FM formation was observed (Fig. 3r, 14.7%, *n* = 1 36). Female fertility of the *carf17* mutant was reduced 23.2% (Supplementary Fig. 3a, e), further suggesting that the *carf17* mutant is abnormal in FM/embryo sac formation. There were 71.1% of *spl-3* ovules with MMC (only 28.9% without MMC, Fig. 3s, *n* = 291 and Fig. 3u); however, 74.2% of *carf17 spl-3* double mutant ovules did not develop MMC (Fig. 3t, *n* = 365 and Fig. 3u). Collectively, our results suggest that *ARF17* is required for promoting MMC specification by genetically acting downstream of *SPL/NZZ*.

## Auxin signaling is involved in MMC specification

To unravel the molecular mechanism by which miR160 and *ARF17* determine the MMC fate, we tested the potential role of auxin signaling in MMC specification. We first examined whether polar auxin transport (PAT) affects MMC formation. Compared with the WT (Fig. 4a, n, 95.2%, *n* = 250), ovules from the *pin1-5* mutant[41] produced two or more MMCLs (Fig. 4b, 35.0%, *n* = 160, Fig. 4c, 33.1%, *n* = 142, Fig. 4n, and Supplementary Fig. 5a–c). Abnormal embryo sacs were also found in *pin1-5* ovules (Supplementary Fig. 5d–h). The PAT inhibitor N-1-Naphthylphthalamic Acid (NPA) inhibits PIN auxin transporters[42], thus we continuously applied NPA to WT inflorescences every 24 h for 4 days, which resulted in supernumerary MMCLs (Fig. 4d, *n*, 62.6%, *n* = 380, Fig. 4e, 15.7%, *n* = 159, Fig. 4f, 42.8%, *n* = 159, and Supplementary Fig. 5i–p) and multiple embryo sacs at different stages per ovule (Supplementary Fig. 5q–x). Callose deposition analysis indicated that multiple MMCLs acquired the MMC identity (Fig. 4g, h, 48.5%, *n* = 202 and Fig. 4n). Furthermore, we ectopically expressed the auxin biosynthesis gene *YUC1*[43] driven by the *EMS1* promoter which is active in the epidermis of nucellus and chalaza[44]. *pEMS1::YUC1* led to extra MMCLs (Fig. 4i, 43.9%, *n* = 255, Fig. 4j, 27.9%, *n* = 136, Fig. 4k, 13.2%, *n* = 136, and Fig. 4n) in 10 examined independent transgenic lines. We also observed multiple MMCs accumulating callose (Fig. 4l, m, 29.9%, *n* = 221, and Fig. 4n). Hence, disruption of PIN-dependent PAT and increased local auxin biosynthesis led to ectopic MMC formation.

To further characterize the role of auxin signaling in MMC specification, we first investigated auxin response using the auxin response/output marker *DR5rev::GFP*[45] when MMC differentiation is normal or abnormal. We found that the *DR5rev::GFP* expression represented a single auxin maximum at the apex of the ovule primordium and nucellus (Supplementary Fig. 6). The auxin maximum is restricted to one cell of the epidermal layer at stages 1-I (Supplementary Fig. 6a) and 1-II (Supplementary Fig. 6b) when the MMCP is present (Supplementary Fig. 1a, b). At stages 2-I (Supplementary Fig. 6c), 2-II (Supplementary Fig. 6d), and 2-III (Supplementary Fig. 6e) when the MMC becomes distinct (Supplementary Fig. 1c–e), the auxin maximum occupies two or three epidermal cells. After the 2-day NPA treatment, ovules produced supernumerary MMCLs (44.6%, *n* = 260) and the auxin maximum changed in terms of numbers and positions in most of ovules at stage 2-III (Fig. 5a–d). We summarized these alterations into three categories, which are Category I: expanded apical maxima—auxin maxima are expanded from the apex to underneath cells (Fig. 5b, 30.2%, *n* = 222 and Fig. 5o), Category II: centrally shifted maxima—auxin maxima are relocated inside nucellus (Fig. 5c, 18.9%, *n* = 222 and Fig. 5o), and Category III: basally shifted maxima—auxin maxima are moved to the chalaza (Fig. 5d, 12.6%, *n* = 222 and Fig. 5o). The 4-day NPA treatment significantly increased the percentage of auxin maximum change in the Category III (Fig. 5o). In addition, similar pattern changes of auxin maxima were also observed in *pEMS1::YUC1* ovules (Supplementary Fig. 7). We then examined auxin accumulation in ovules using the auxin reporter R2D2[46]. A high level of auxin was detected in nucellus, including the MMC, at stages 2-II (Fig. 5e) and 2-III (Fig. 5f). NPA treatment caused auxin accumulation in both nucellus and chalaza at stages 2-II (Fig. 5g) and 2-III (Fig. 5h).

We also examined the PIN1 expression domain after NPA treatment during ovule development. In *pPIN1::PIN1-GFP*[47] ovules, PIN1 is mainly found in epidermal cells of nucellus at stage 1-I (Supplementary Fig. 8a), then also appeared in one file of cells in the center of chalaza at stage 1-II (Supplementary Fig. 8b). At stages 2-I (Fig. 5i and Supplementary Fig. 8c), 2-II (Fig. 5k and Supplementary Fig. 8d), and 2-III (Fig. 5m and Supplementary Fig. 8e), PIN1 is present in epidermal cells of nucellus, integuments, and in a few files of cells in the central chalaza. The NPA treatment led to expansions of PIN1 expression domain in the central cells of chalaza at stages 2-I (Fig. 5j), 2-II (Fig. 5l), and 2-III (Fig. 5n). In summary, our findings suggest that the spatially restricted auxin activity mediated by PIN1 is important for MMC fate acquisition.

## miR160 and *ARF17* control establishment of the local auxin maximum via defining the expression domain of PIN1

To gain insight into the possible relationship between the miR160-regulated *ARF17* and the local auxin activity in specification of MMC, we examined the potential effects of miR160 and *ARF17* on auxin response and accumulation. A single auxin maximum was present at the apex of nucellus in *DR5rev::GFP* ovules at stage 2-III (Fig. 6a); however, *DR5rev::GFP foc* ovules showed alterations of auxin maxima in three categories (Fig. 6b–d, q), resembling that NPA-treated ovules (Fig. 5b–d). Similar changes of auxin maxima were also observed in *DR5rev::GFP pARF17::mARF17* (Fig. 6e, q) and *DR5rev::GFP pARF17::mARF17 foc* (Fig. 6f, q) ovules. Examination of R2D2 expression found

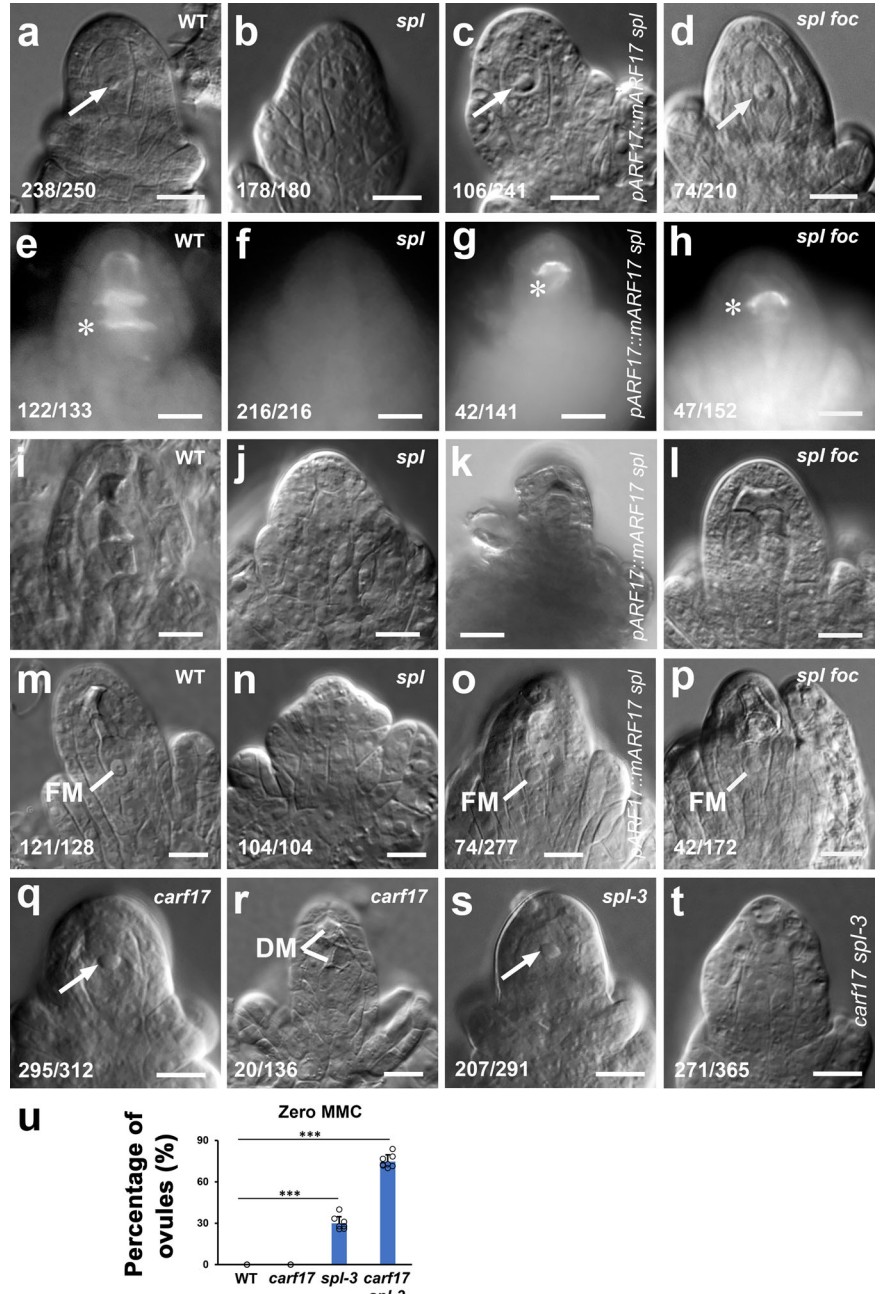

**Fig. 3 | *ARF17* controls MMC specification. a–l** Overexpression of *ARF17* in the *spl* mutant rescued the MMC formation. DIC images showing one MMC in WT (**a**, arrow), no MMC in *spl* (**b**), one MMCL in *pARF17::mARF17 spl* (**c**, arrow), and one MMCL in *spl foc* (**d**, arrow) ovules. **e–l** Callose deposition (denoted by white asterisks) displaying one MMC undergoing meiosis in WT (**e**), *pARF17::mARF17 spl* (**g**), and *spl foc* (**h**) ovules, but no meiosis occurrence in the *spl* ovule (**f**). **e–h** Callose staining. **i–l** DIC image of **e–h**. **m–p** Overexpression of *ARF17* in the *spl* mutant rescued the functional megaspore (FM) formation. DIC images showing FMs in WT (**m**), *ARF17::mARF17 spl* (**o**), and *spl foc* (**p**) ovules, but no FM in the *spl* ovule (**n**).

**q–t** Requirement of *ARF17* for MMC specification. DIC images showing an MMC in the *carf17* ovule (**q**, arrow) but the abnormal megasporogenesis indicated by all degenerated megaspores (DM, **r**). DIC images showing an MMC in the *spl-3* (**s**, arrow) ovule, but no MMC in the *carf17 spl-3* ovule (**t**). **u** Quantifications of zero MMC in WT, *carf17*, *spl-3*, and *carf17 spl-3* ovules from 8 individual plants (*n* = 8 plants; two-sided Student's *t* test; Error bars: SD; \**p* < 0.05, \*\**p* < 0.01, and \*\*\**p* < 0.001). Source data are provided as a Source data file. Numbers in the panels denote frequencies of phenotypes shown. Scale bars, 10 μm.

that more auxin was accumulated in nucellus in *foc* ovules at stages 2-III (Fig. 6g, h).

We then tested whether miR160 and *ARF17* regulate the PIN1 expression domain. Compared to its expression in *pPIN1::PIN1-GFP* ovules (Fig. 6i, m and Supplementary Fig. 9a), PIN1 expression domains were expanded in the central chalaza of *pPIN1::PIN1-GFP foc* (Fig. 6j, n and Supplementary Fig. 9b), *pPIN1::PIN1-GFP pARF17::mARF17* (Fig. 6k, o and Supplementary Fig. 9c), and *pPIN1::PIN1-GFP pARF17::mARF17 foc* (Fig. 6l, p and Supplementary Fig. 9d) ovules at stages 1-II, 2-I, and 2-III.

Our results suggest that Moreover, *ARF17* genetically interacts domains of PIN1, which is critical for establishment of a single local auxin maximum at the ovule apex and accumulation of auxin in nucellus, including the MMC.

### Auxin signaling specifies MMC through *ARF17* and *SPL/NZZ*

We finally tested whether auxin signaling controls MMC specification through *ARF17* and *SPL/NZZ*. The auxin maximum was almost undetectable at the apex of *DR5rev::GFP spl* ovule where the MMC is not

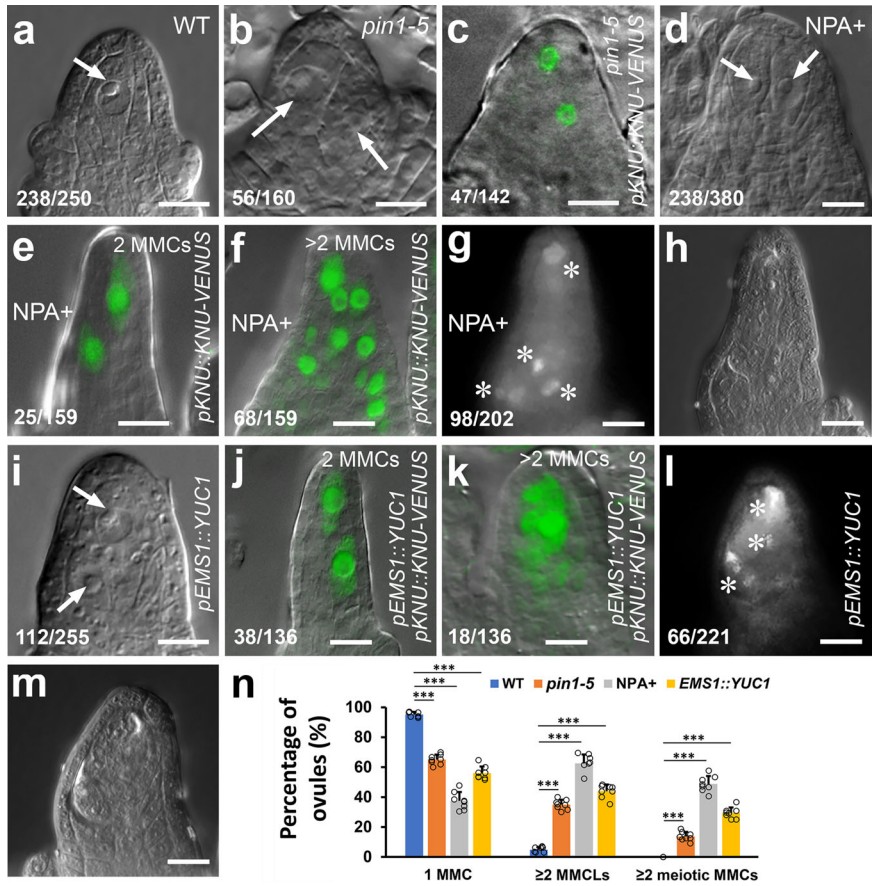

**Fig. 4 | Manipulation of auxin signaling alters MMC specification. a–c** Loss-of-function of *PIN1* causing supernumerary MMCLs. DIC images showing one MMC (arrow) in WT (**a**) and two MMCLs (arrows) in *pin1-5* (**b**) ovules. **c** A merged confocal and DIC image of the *pin1-5* ovule expressing *pKNU::KNU-VENUS* showing two MMCLs. **d–h** Inhibiting polar auxin transport by NPA resulting in supernumerary MMCLs and MMCs. **d** A DIC image showing two MMCLs (arrows) in the 4-day NPA-treated WT ovule. **e, f** Merged confocal and DIC images of 4-day NPA-treated ovules expressing *pKNU::KNU-VENUS* marking the MMC fate. **e** 2 MMCLs. **f** >2 MMCLs. **g, h** Callose deposition (denoted by white asterisks) showing multiple MMCs undergoing meiosis in the NPA-treated ovule. **g** Callose staining. **h** DIC image of **g**.

**i–m** Alteration of auxin biosynthesis leading to supernumerary MMCLs and MMCs. **i** A DIC image showing two MMCLs (arrows) in the *pEMS1::YUC1* ovule. **j, k** Merged confocal and DIC images of *pEMS1::YUC1* ovules expressing *pKNU::KNU-VENUS* marking the MMC fate. **j** 2 MMCLs. **k** >2 MMCLs. **l, m** Callose deposition (denoted by white asterisks) showing multiple MMCs undergoing meiosis in the *pEMS1::YUC1* ovule. **l** Callose staining. **m** DIC image of **l**. **n** Quantifications of MMCs and MMCLs in WT, *pin1-5*, NPA-treated, and *pEMS1::YUC1* ovules from 8 individual plants ($n = 8$ plants; two-sided Student's *t* test; Error bars: SD; *$p < 0.05$, **$p < 0.01$, and ***$p < 0.001$). Source data are provided as a Source data file. Numbers in the panels denote frequencies of phenotypes shown. Scale bars, 10 μm.

formed (Fig. 7a, 100%, $n = 149$); however, in the *DR5rev::GFP pARF17::mARF17 spl* ovule with the MMC, the auxin maximum was restored to the ovule apex (Fig. 7b, 33.1%, $n = 133$). As described previously[48], PIN1 was almost not observed either in the epidermis of nucellus or the central chalaza in the *pPIN1::PIN1-GFP spl* ovule (Fig. 7c, 94.3%, $n = 70$), while the PIN1 expression was recovered in the *pPIN1::PIN1-GFP pARF17::mARF17 spl* ovule (Fig. 7d, 34.2%, $n = 146$). Our results suggest that miR160-regulated *ARF17* is required for attaining the PIN1-established auxin maximum at the ovule apex. After a 4-day NPA treatment, 62.6% of WT ovules produced supernumerary MMCLs (Fig. 4d, $n = 380$ and Fig. 7p), but 36.1% of *carf17* (Fig. 7e, p, $n = 294$) and 15.2% of *spl-3* (Fig. 7p, $n = 344$) ovules formed extra MMCLs. MMC was not found in 74.2% of *carf17 spl-3* double mutant ovules (Fig. 3t, $n = 365$). NPA treatment did not induce the formation of MMC in 74.9% of *carf17 spl-3* ovules and 25.1% ovules only formed one MMC (Fig. 7f, p, $n = 306$). Thus, the failure of supernumerary MMCL production in NPA-treated *carf17 spl-3* ovules suggests that the auxin signaling-induced MMC specification requires *ARF17* and *SPL/NZZ*.

To examine whether auxin signaling regulates expression of *MIR160a* (*FOC*), miR160, and *ARF17*, we treated *pMIR160a5′::NSL-3xGFP::MIR160a3′*[32], *pUBI10::miR160sensor-NSL-3xGFP*, and *pARF17::ARF17-GFP* plants with NPA. The expression of *MIR160a* gene was

decreased after 1-day and 4-day NPA treatments (Fig. 7g–i), and NPA treatment decreased accumulation of the mature miR160 in the MMC and other cells in nucellus and chalaza (Fig. 7j–l). Conversely, 1-day and 4-day NPA treatments enhanced the expression of ARF17 protein in subepidermal nucellus cells, especially in MMCLs (Fig. 7m–o), suggesting that the formation of MMC requires ARF17. Furthermore, qRT-PCR results showed the expression of *ARF17* was decreased in the *spl* ovule, and the expression of *SPL/NZZ* was also decreased in the *carf17* ovule (Fig. 7q). The expression of *PIN1* was decreased in the *carf17* ovule but increased in the *pARF17::mARF17* (Fig. 7q) ovule. In summary, our results suggest that auxin signaling modulated by the miR160-targeted *ARF17*, *SPL/NZZ*, and PIN1 provides the spatially restricted information for the proper specification of a single MMC per ovule.

## Discussion

In many animals, germline cells are differentiated and segregated from soma during early embryogenesis, whereas flowering plants generate male and female germline cells post-embryonically in a flower. Previous studies have reported that several pathways involved in cell cycle control, signal transduction, ta-siRNAs, and epigenetic regulation restrain the number of MMCs, i.e., the female germline cells[5–14,16]. Here we report that the miRNA-controlled, spatially restricted auxin

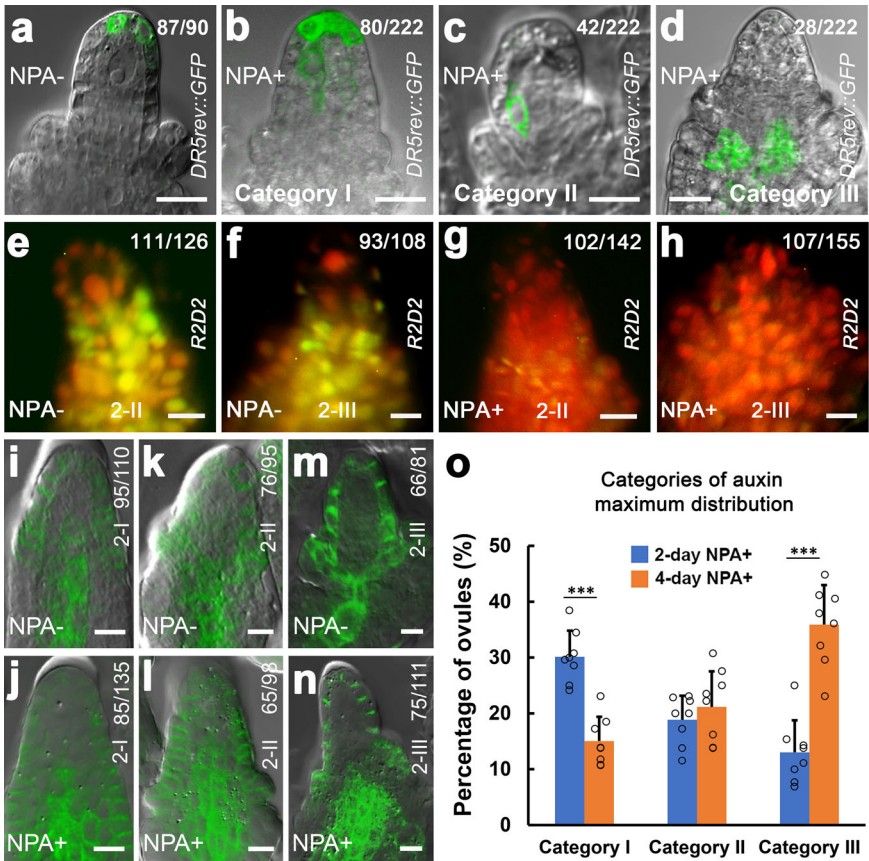

**Fig. 5 | Establishment of local auxin signaling in the ovule requires PIN1.**
**a**–**d** Merged confocal and DIC images of ovules expressing *DR5rev::GFP* showing three categories of auxin maximum changes after 2-day NPA treatment at stage 2-III. **a** One auxin maximum at the peak of nucellus without NPA treatment (NPA-). **b**–**d** NPA treatment (NPA+). **b** Category I: expanded apical maxima. **c** Category II: centrally shifted maxima. **d** Category III: basally shifted maxima. **e**–**h** Accumulation of auxin in ovules. Fluorescence images of ovules expressing R2D2 showing mDII-ntdTomato (red), DII-n3×Venus (green), and overlayed fluorescence signal (yellow). Red color indicates accumulation of auxin. High level of auxin (red) in the nucellus including the MMC without NPA treatment (NPA−) at stages 2-II (**e**) and 2-III (**f**). More accumulation of auxin (red) in the nucellus and chalaza with NPA treatment

(NPA+) at stages 2-II (**g**) and 2-III (**h**). **i**–**n** Merged confocal and DIC images of ovules expressing *pPIN1::PIN1-GFP* showing the PIN1 expression in epidermal cells of nucellus and the central chalaza at stage 2-I (**i**), in epidermal cells of nucellus, integuments, and the central chalaza at stages 2-II (**k**) and 2-III (**m**) without NPA treatment and the expanded PIN1 expression domain in the chalaza after 2-day NPA treatment at stages 2-I (**j**), 2-II (**l**), and 2-III (**n**). **o** Quantifications of auxin maximum distribution in ovules from eight individual plants at stage 2-III after 2-day and 4-day NPA treatment (*n* = 8 plants; Two-sided Student's *t* test; Error bars: SD; *$p < 0.05$, **$p < 0.01$, and ***$p < 0.001$). Source data are provided as a Source data file. Numbers in the panels denote frequencies of phenotypes shown. Scale bars, 10 μm.

signaling promotes the specification of one MMC per ovule in *Arabidopsis* (Fig. 7r, s). During the normal ovule development, the *MIR160a* gene is expressed in ovule cells, while the mature miR160 is particularly active in a single hypodermal cell, which is underneath the auxin maximum at the apex of ovule (Figs. 2a–i, 5a and 7r). miR160 down-regulates the expression of *ARF17* (Fig. 2c, d), whereas auxin induces the expression of *ARF17*[32]. The ARF17 protein is mainly found in the MMC (Fig. 2m–p). Auxin present in the MMC might activate ARF17 (Fig. 5e, f). Moreover, *ARF17* genetically interacts with *SPL/NZZ* (Fig. 3), and they possibly affect each other's expression (Fig. 7q). ARF17 and *SPL*/NZZ also affect the *PIN1* expression domain (Figs. 6i–p and 7c, d), which may contribute to the establishment of auxin maximum at the apex of ovule and accumulation of auxin in the MMC (Fig. 5a–h). Thus, a delicate balance between miR160 and auxin signaling leads to a precise control of ARF17 function, which promotes one hypodermal cell to acquire the MMC identity (Fig. 7r, s).

*SPL/NZZ* and *ARF3* are important for MMC differentiation, as the *spl/nzz* mutant fails to form MMC[15,17] and ectopic expression of *ARF3* results in extra MMCLs[12]. The MADS-box transcription factor STK (SEEDSTICK) upregulates expression of *AGO9*, *RDR6*, and *DRM*, while AGO9, RDR6, and DRM epigenetically repress the expression of *SPL/NZZ*[4]. Ectopic expression of *SPL/NZZ* in ovules of *stk*, *drm1 drm2*,

*ago9-2*, *rdr6-11* mutants, and the *35S::SPL/NZZ* plant leads to the formation of supernumerary MMCL, but only one of these MMCLs expresses the MMC marker *KNU* and undergoes meiosis. Furthermore, both the *SPL*/NZZ transcript and the *SPL*/NZZ protein are restricted to only a few cells of nucellus epidermis, but they are not present in the MMC or its precursor cell. These results suggest that *SPL/NZZ* is not sufficient for the complete differentiation of MMC, although it is required for the initial MMC specification non-cell autonomously. *SPL/NZZ* was suggested to be involved in auxin homeostasis[49]. A recent study showed that auxin distribution is associated with differentiation of the male germ cell PMC (pollen mother cell)[50]. Mutations in auxin biosynthesis genes *TAA1* (*TRYPTOPHAN AMINOTRANSFERASE OF ARABIDOPSIS 1*) and *TAR2* (*TRYPTOPHAN AMINOTRANSFERASE RELATED 2*) impair the PMC formation, whereas ectopic expression of *SPL/NZZ* partially rescues the PMC specification. Different from its expression pattern in MMC, *SPL/NZZ* is expressed in PMC, suggesting that *SPL/NZZ* might directly promote the PMC differentiation. Nevertheless, the elevated expression of *SPL/NZZ* does not result in excess PMC but the loss of PMC identity in anthers. tasiR-ARFs maintain the expression of *ARF3* at the basal end of chalaza via suppressing its expression in apical and hypodermal MMC neighbor cells, which may prevent them from acquiring the MMC fate[12]. Ectopic expression of the

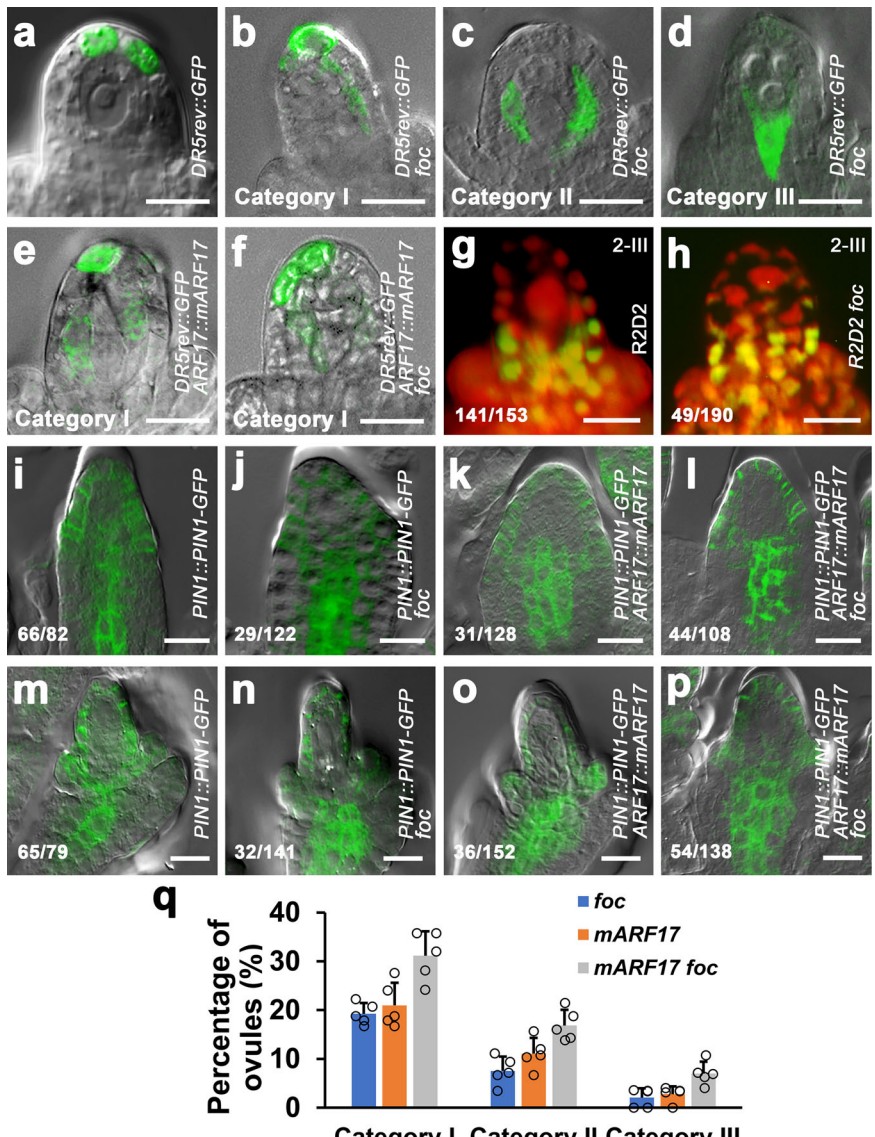

**Fig. 6 | miR160 and *ARF17* control the local auxin maximum in ovule via defining the PIN1 expression domain. a–f** Merged confocal and DIC images of ovules expressing *DR5rev::GFP* showing auxin maximum distributions. **a** One auxin maximum at the apex of nucellus of the *DR5rev::GFP* ovule. *DR5rev::GFP foc* ovules showing changes of auxin maxima in Category I (**b**, expanded apical maxima), Category II (**c**, centrally shifted maxima), and Category III (**d**, basally shifted maxima). Auxin maxima in Category I from *DR5rev::GFP pARF17::mARF17* (**e**) and *DR5rev::GFP pARF17::mARF17 foc* (**f**) ovules. **g**, **h** Changes of auxin accumulation in the *foc* ovule. Fluorescence images of ovules expressing R2D2 showing a high level of auxin (red) in the nucellus, especially in the MMC of the WT ovule at stage 2-III (**g**) but auxin

accumulation in extra MMCLs in the nucellus of the *foc* ovule at stage 2-III (**h**). Merged confocal and DIC images of ovules showing the effect of *foc* and *ARF17* on PIN1 expression domain at stages 2-I (**i–l**) and 2-III (**m–p**). **i**, **m** PIN1 is present at the nucellus epidermis, the center of chalaza, and integuments in *pPIN1::PIN1-GFP* ovules. The PIN1 domain is expanded in chalaza in *pPIN1::PIN1-GFP foc* (**j**, **n**), *pPIN1::PIN1-GFP pARF17::mARF17* (**k**, **o**), and *pPIN1::PIN1-GFP pARF17::mARF17 foc* (**l**, **p**) ovules.
**q** Quantifications of auxin maximum distribution in *DR5rev::GFP foc*, *DR5rev::GFP pARF17::mARF17*, and *DR5rev::GFP pARF17::mARF17 foc* ovules from five individual plants (*n* = 5 plants; Error bars: SD). Source data are provided as a Source data file. Numbers in the panels denote frequencies of phenotypes shown. Scale bars, 10 μm.

ARF3 protein in the subepidermal cells surrounding the MMC causes supernumerary MMCLs. However, the fact that *ARF3* is not expressed in the MMC suggests that *ARF3* does not directly specify MMC.

A complex auxin signaling pathway which involves *ARF17*, *SPL/NZZ*, and *PIN1* is required for MMC specification. The auxin maximum is restricted to one to three cells in the epidermal layer at the apex of nucellus with the progress of MMC differentiation. One hypodermal cell underneath the auxin maximum enlarges and eventually differentiate into MMC. In the *spl* ovule which fails to develop the MMC, the auxin maximum is not formed at the tip of nucellus (Fig. 7a), and PIN1 is not present in the epidermal cells of nucellus, integuments, and the central chalaza (Figs. 5i–m and 7c). Overexpression of *ARF17* can restore the auxin maximum to the

nucellus apex by rescuing the normal PIN1 expression in the *spl* ovule (Fig. 7a–d), suggesting that auxin signaling modulated by *ARF17*, *SPL/NZZ*, and PIN1 are important for MMC differentiation. In the apomictic *Hieracium* subgenus *Pilosella* ovule, NPA treatment increases the expression level of *DR5::GFP* and alters the *DR5::GFP* expression domain[51]. However, NPA treatment does not affect the MMC differentiation, although the number of aposporous initial cells is increased, suggesting that auxin signaling might play a more dominant role in the apomixis process in apomictic species.

We found that the formation of supernumerary MMCLs in *foc*, *pARF17::mARF17*, *pARF17::mARF17 foc*, *pEMS1::YUC1*, *pin1-5*, and NPA-treated ovules is associated with location changes of auxin maxima. These MMCLs express the MMC marker *KNU* and a portion of them

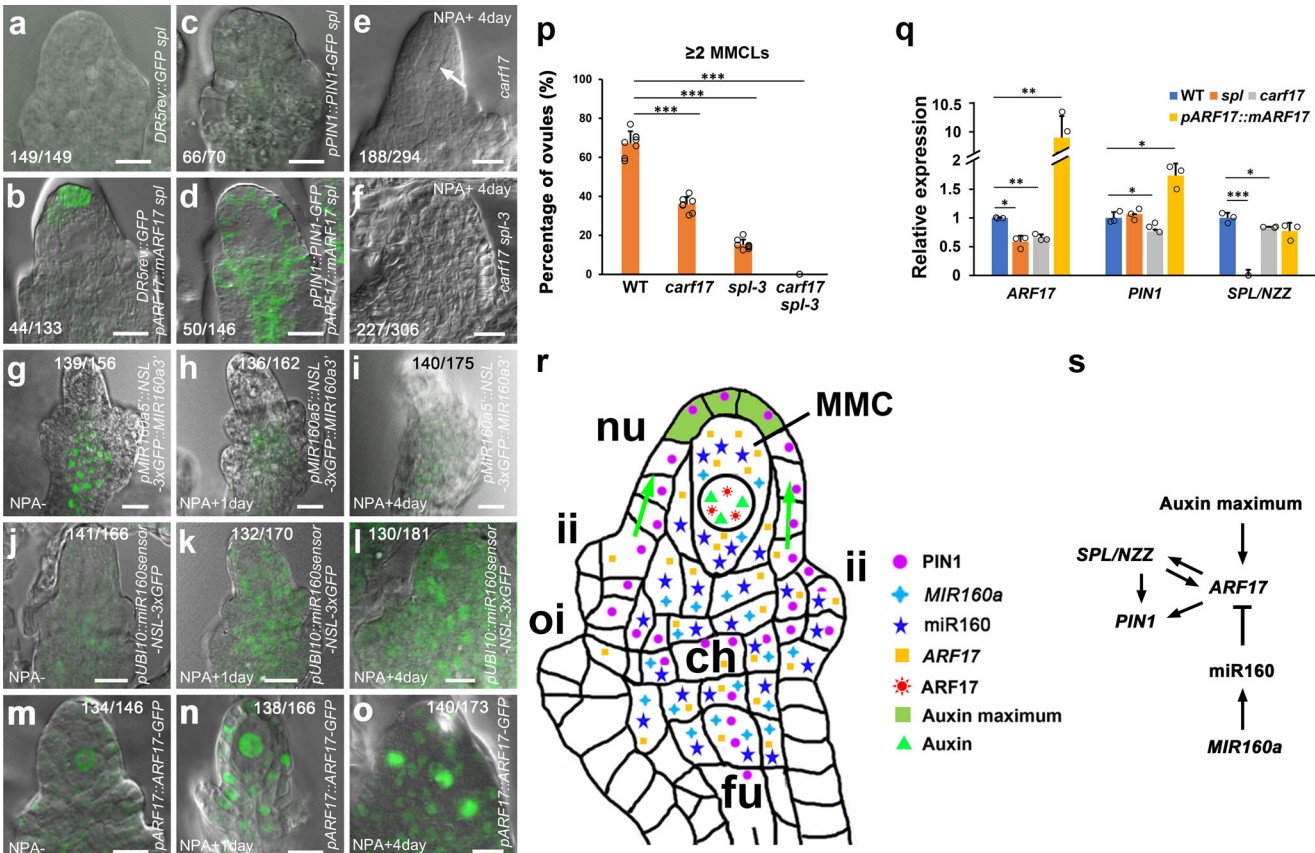

**Fig. 7 | Auxin signaling controls MMC specification through *ARF17* and *SPL/NZZ*. a–d** Merged confocal and DIC images of ovules showing rescued auxin signaling by overexpression of *ARF17*. **a** Absence of auxin maximum in the *DR5rev::GFP spl* ovule. **b** The restored auxin maximum to the apex of nucellus in the *DR5rev::GFP pARF17::mARF17 spl* ovule. **c** Almost no detectable PIN1 in the *pPIN1::PIN1-GFP spl* ovule. **d** Partially rescued PIN1 expression at the epidermis of nucellus and chalaza in the *pPIN1::PIN1-GFP pARF17::mARF17 spl* ovule. **e, f** DIC images of ovules showing one MMC (arrow) in the *carf17* ovule, but no MMC in the *carf17 spl-3* ovule after NPA treatment. **g–i** NPA treatment represses the *MIR160a* (*FOC*) expression. Merged confocal and DIC images of *pMIR160a5'::NSL-3xGFP::MIR160a3'* ovules showing the decreased expression of *MIR160a* in chalaza after 1-day (**h**) and 4-day (**i**) NPA treatment comparing with the control (**g**). **j–l** NPA treatment represses the miR160 accumulation. Merged confocal and DIC images of *pUB-I10::miR160sensor-NSL-3xGFP* ovules showing increased GFP signals in nucellus, particularly the MMC, and chalaza after 1-day (**k**) and 4-day (**l**) NPA treatment comparing with the control (**j**). **m–o** NPA treatment increases the ARF17 protein expression in MMCLs. Merged confocal and DIC images of *pARF17::ARF17-GFP* ovules showing elevated levels of ARF17 protein in MMCLs and other subepidermal

nucellus cells after 1-day (**n**) and 4-day (**o**) NPA treatment comparing with the control (**m**). **p** Quantifications of ≥2 MMCLs in WT, *carf17*, *spl-3*, and *carf17 spl-3* ovules with NPA treatment from eight individual plants (*n* = 8 plants; two-sided Student's *t* test; Error bars: SD; *$p < 0.05$, **$p < 0.01$, and ***$p < 0.001$). **q** qRT-PCR results showing expression of *ARF17*, *PIN1*, and *SPL/NZZ* in WT, *spl*, *carf17* and *pARF17::mARF17* ovules. Values were normalized as relative expression to *ACTIN2*. (*n* = 3 biological replicates; two-sided Student's *t* test; Error bars: SD; *$p < 0.05$, **$p < 0.01$, and ***$p < 0.001$). Source data are provided as a Source data file. **r** A model illustrating the MMC specification controlled by miR160-orchestrated auxin signaling. Green arrows indicate auxin flow. Colored shapes indicate presence of the mature miR160, auxin, mRNAs, and proteins. ch chalaza, fu funiculus, ii inner integument, MMC megaspore mother cell, nu nucellus, and oi outer integument. **s** The proposed regulatory network associated with auxin signaling, *MIR160a*, miR160, *ARF17*, *PIN1*, and *SPL/NZZ* during the MMC specification. Arrows indicate the positive regulation/interaction, while the T-bar indicates the negative regulation/interaction. Numbers in the panels denote frequencies of phenotypes shown. Scale bars, 10 μm.

accumulate callose, suggesting that they are preparing to undergo meiosis. Not only in the distal end of nucellus, MMCLs were also observed in other nucellus region (Figs. 1i and 4f, g, k, l and Supplementary Fig. 5o, p). We also observed the ARF17 protein primarily in the normally differentiated MMC as well as in supernumerary MMCLs caused by the ectopic expression of *ARF17* (Fig. 2m–p) and by the alteration of auxin signaling (Fig. 7m–o). The acquisition of MMC identity depends on ARF17 (Figs. 3u and 7p); however, overexpression of *ARF17* specifically in the MMCP/MMC does not result in the formation of supernumerary MMCLs (Fig. 2q–x). Therefore, different from previously studied regulators[5–22], our results suggest that *ARF17* is a key determinant for promoting the MMC specification instead of the MMC proliferation. Collectively, we uncover a molecular mechanism underlying the MMC fate determination, which thus lays the foundation to female germline in plants.

## Methods

### Plant materials

*Arabidopsis thaliana* Landsberg *erecta* (L*er*) was used as the wild type (WT) unless otherwise noted. The mutants, marker lines, and transgenic lines used in this study were *foc*[32], *spl*[15,17], *DR5rev::GFP*[45], *pKNU::KNU-VENUS*[11], *pin1-5*[41], *pPIN1::PIN1-GFP*[47], *pPIN1::PIN1-GFP spl*[48], *spl-3** (SALK_090804, * indicated generated/identified in this study), *pAT 5G01860::n1GFP** (the female gametophyte marker)[52], *pAT5G4 5980::n1GFP** (the egg cell marker)[52], *pAT5G05490::n1GFP** (the central cell marker)[52], *pAT5G56200::n1GFP** (the antipodal cell marker)[52], *pARF17::ARF17*, *pARF10::mARF10**, *pARF16::mARF16**, *pARF17::-mARF17**, *pARF17::mARF17 foc**, *pKNU::KNU-VENUS foc**, *pKNU::KNU-VENUS pARF17::mARF17**, *pKNU::KNU-VENUS pARF17::mARF17 foc**, *pKNU::mARF17**, *pKNU::mARF17 pKNU::KNU-VENUS**, *pKNU::STTM160/160-48**, *pKNU::STTM160/160-48 pKNU::KNU-VENUS**, *pARF17::ARF17-*

*GFP\**, *pARF17::mARF17-GFP\**, *pMIR160a5'::NSL-3xGFP::MIR160a3\**, *pUBI10::NSL-3xGFP\**, *pUBI10::miR160sensor-NSL-3xGFP\**, *pUBI10::NSL-3xGFP foc\**, *pUBI10::miR160sensor-NSL-3xGFP foc\**, *pEMS1::YUC1\**, *pKNU::KNU-VENUS pEMS1::YUC1\**, *DR5rev::GFP pEMS1::YUC1\**, *crispr-arf17* (*carf17*)*, *spl-3 carf17\**, *DR5rev::GFP foc\**, *DR5rev::GFP pARF17::mARF17\**, *DR5rev::GFP pARF17::mARF17 foc\**, *pARF17::mARF17 spl\**, *foc spl\**, *DR5rev::GFP spl\**, *DR5rev::GFP pARF17::mARF17 spl\**, *pPIN1::PIN1-GFP foc\**, *pPIN1::PIN1-GFP pARF17::mARF17\**, *pPIN1::PIN1-GFP pARF17::mARF17 foc\**, *pPIN1::PIN1-GFP pARF17::mARF17 spl\**, *R2D2* (auxin accumulation marker)[46], and *R2D2 foc\**. These plants were either in the L*er* background or crossed into the L*er* background five times before examination.

## Plant growth conditions

All plants were grown in Metro-Mix 360 soil (Sun-Gro Horticulture City State) in growth chambers under a 16-h light/8-h dark photoperiod, at 22 °C and 50% of humidity.

## Generation of constructs and transgenic plants

For genetic analyses, gene expression, and protein localization studies, promoters of *ARF10*, *ARF16*, and *ARF17*[34] were cloned into the pENTR/D-TOPO vector (Invitrogen). Site mutations of *ARF10*[35], *ARF16*[34], and *ARF17*[33] cDNAs were generated by overlapping PCR to produce miR160-resistant versions of *mARF10*, *mARF10*, and *mARF17*. The *mARF10*, *mARF16*, and *mARF17* were cloned into pENTR/D-TOPO vectors containing corresponding promoters to generate *pENTR-pARF10::mARF10*, *pENTR-pARF16::mARF16*, and *pENTR-pARF17::mARF17*, respectively.

The *MIR160a* 5' promoter[32] was PCR-amplified from the BAC clone T16B24 with KpnI and XbaI digestion sites and then cloned into pENTR/D-TOPO vector to generate *pENTR-pMIR160a5'*. An NSL-3xGFP fragment was digested by KpnI and XbaI from *pGreenII KAN SV40-3×GFP*[53] and inserted into the *pENTR-pMIR160a5'* to produce the *pENTR-pMIR160a5'::NSL-3xGFP* vector. The PCR-amplified *MIR160a3'* fragment was then inserted into *pENTR-pMIR160a5'::NSL-3xGFP* at XbaI and AscI sites to generate *pENTR-pMIR160a5'::NSL-3xGFP::MIR160a3'*.

To generate the nucleus-localized GFP-based miR160 sensor construct, a 634 bp genomic DNA fragment of the *UBI10* promoter was first PCR-amplified from the genomic DNA of L*er*[39]. *pMIR160a5'* was replaced by *pUBI10* through SacII + KpnI sites in the *pENTR-pMIR160a5'::NSL-3xGFP* vector to generate *pENTR-pUBI10::NSL-3xGFP*. The *UBI10* promoter with the sensor sequence (TGGCATGCAGGGAGCCAGGCA)[38] at the 3' end was PCR-amplified and the resulting fragment was used to replace *pMIR160a5'* in the *pENTR-pMIR160a5'::NSL-3xGFP* vector to produce *pENTR-pUBI10::miR160sensor-NSL-3xGFP*.

The PCR-amplified 2 kb *KNU* promoter[11] was amplified from the genomic DNA of L*er* and cloned into the pENTR/D-TOPO vector to generate *pENTR-pKNU*. The *mARF17* was PCR-amplified as described above and subsequently cloned into *pENTR-pKNU* to produce *pENTR-pKNU::mARF17*. The fragment containing the sequence of an STTM (Short Tandem Target Mimic) targeting the miR160 was PCR-amplified from the STTM160/160-48[40] construct and then cloned into *pENTR-pKNU* to generate *pENTR-pKNU::STTM160/160-48*.

The PCR-amplified 1.7 kb *EMS1* promoter was cloned into the pENTR/D-TOPO vector to generate *pENTR-pEMS1*[44]. The *YUC1* cDNA was PCR-amplified from the pCHF3 plasmid[43] and subsequently cloned in the *pENTR-EMS1* vector to produce *pENTR-pEMS1::YUC1*.

To generate *crispr-arf17* (*carf17*) lines, the egg cell-specific promoter-controlled Cas9 vector pHEE401E was used in this study[54]. Target sites of 23-bp sequences were searched using CRISPR-PLANT (https://www.genome.arizona.edu/crispr/CRISPRsearch.html). A pair of synthesized 23-nt oligoes (100 μmol/l each) containing the target site within the first exon and the BsaI digestion site were heated at 95 °C for 5 min and annealed at the room temperature. Then the Golden Gate reaction was conducted with the annealed insert, the

pHEE401E vector, BsaI-HF®v2 (NEB #R3733), and the T4 DNA Ligase (NEB # M0202T) to produce the *crispr-arf17* construct.

The pGWB1 binary vector was used for genetic and expression analyses, while the pGWB4 binary vector harboring the *GFP* gene was employed for protein localization studies[55]. The final constructs *pARF10::mARF10*, *pARF16::mARF16*, *pARF17::mARF17*, *pKNU::mARF17*, *pKNU::STTM160/160-48*, *pARF17::ARF17-GFP*, *pARF17::mARF17-GFP*, *pMIR160a5'::NSL-3xGFP::MIR160a3'*, *pUBI10::NSL-3xGFP*, *pUBI10::miR160sensor-NSL-3xGFP*, and *pEMS1::YUC1* were generated using the Gateway LR Recombinase II Enzyme Mix (Invitrogen). Detailed information for all constructs and primers is shown in Supplementary Tables 2 and 3. The resulting vectors were transformed into *Agrobacterium* strain GV3101 and plant transformation was performed using the floral dip method[56]. The transformants were screened on ½ Murashige and Skoog (MS) plates containing 25 μg/ml hygromycin.

## NPA treatment

The 50 mM of auxin transport inhibitor N-1-Naphthylphthalamic Acid (NPA) (Sigma; N-12507-250MG) stock solution was prepared in DMSO[57]. The working NPA solution was 50 μM in 0.01% of Silwet L-77 (Lehle Seeds, Round Rock, TX, USA). The solution containing 0.1% of DMSO and 0.01% of Silwet L-77 was used for mock treatment. One drop of NPA solution was applied to the top of the main inflorescence at 10 a.m. every day. Samples were collected at 2 days and 4 days for analyses.

## Whole-mount RNA in situ hybridization and qRT-PCR

RNA in situ hybridization was carried out in whole-mount ovules[58]. The probes of *MIR160a* and *ARF17* were described in our previous studies[32]. A miRCURY LNA™ Detection probe with the DIG oligonucleotide 3-end labeling (5'-TGGCATACAGGGAGCCAGGCA-3') was synthesized (Exiqon) for examining the expression of mature miR160[59]. Briefly, pistils were removed from flowers and opened along one side of replum. Opened pistils were fixed with vacuum for 1 h. After a series of processes, including chlorophyll removal, permeabilization, and hybridization, ovules were dissected out from pistils for signal detection using Western Blue® (Promoga). Images were taken via an Olympus BX51 microscope equipped with an Olympus DP 70 digital camera.

For quantitative reverse transcription PCR (qRT-PCR)[32,60,61], ovules at stages 2-II and 2-III were collected under dissection microscope from WT, *spl*, *carf17*, and *pARF17::mARF17* gynoecia. Total RNAs were extracted using the RNeasy Plant Mini Kit (Qiagen). Reverse transcription reactions were then conducted via a QuantiTect Reverse Transcription Kit (Qiagen) after determining the RNA concentration by a NanoDrop ND-1000 spectrophotometer (Thermo Scientific). The quantitative real time PCR (DNA Engine Opticon 2 system) were performed using the Fast SYBR Green PCR Master Mix and primers listed in Table S3. ACTIN2 was used as an internal control. Three biological replicates were performed.

## Whole-mount immunolocalization

For whole-mount immunolocalization, pistils were harvested, dissected under a dissection microscope, and then embedded in the polyacrylamide solution[62]. A coverslip was put on the top of the polyacrylamide solution and a tweezer was used to press the coverslip to push ovules out of the pistils. Ovules were permanently embedded in the polyacrylamide matrix after polymerization at room temperature for at least 20 min. After cell wall digestion and cell membrane permeabilization, ovules were blocked by 1% of BSA for 1 h at 37 °C. The anti-GFP antibody (Torrey Pines Biolabs; catalog no. TP401) was applied by 1:100 dilution overnight. After washing five times each for 15 min, samples applied the secondary antibody Goat Anti-Rabbit IgG Antibody (Alexa Fluor® 488) (Jackson ImmunoResearch Lab; 111-545-003) were incubated overnight. One drop of Prolong Live Antifade

reagent (Life Technologies; P36975) was applied to samples which were washed five times each for 15 min. The samples were then mounted for observation. All steps were carried out at 4 °C unless otherwise noted.

## Phenotypic analyses and microscopy

To examine megaspore mother cell (MMC) differentiation, flower buds were collected and fixed in 37% of methanol stabilized formaldehyde at room temperature with vacuum for 1 h. A graded ethanol series (20% of increments) was carried out for bud clearance. Ovules were dissected out from pistils and mounted in the Hoyer's solution[63] overnight before imaging. Ovules were photographed on a Leica TCS SP2 laser scanning confocal microscope under Differential interference contrast (DIC) optics.

Ovules examined by scanning electron microscopy (SEM) were performed as described in our previous studies[32] using a Hitachi S-570 scanning electron microscope.

The ovule semi-thin section was conducted based on our previous studies[60,61,64]. Images were photographed by an Olympus BX51 microscope through an Olympus DP 70 digital camera.

Callose deposition assay was conducted for examining MMC meiosis[14,16]. Briefly, inflorescences were fixed by FAA for 16 h and washed by 1xPBS three times each for 10 min. Pistils were incubated in 0.1% of aniline blue in 100 mM of Tris (pH 8.5) for 12 h. Ovules were dissected out in 30% of glycerol and observed with an Olympus (BX51) microscope under the UV light (365-nm excitation and 420-nm emission). Three biological replicates were performed.

Ovules from plants expressed R2D2 were dissected and mounted in 10% of glycerol. Fluorescence filters for TRITC and FITC were used to acquire signals of tdTomato[65] and VENUS[66], respectively.

For analyzing GFP signals, ovules were mounted in water and observed with a Leica TCS SP2 laser scanning confocal microscope using a ×63/1.4 water immersion objective lens. The 488-nm laser line was used to excite GFP. The PMT gain settings was held at 650. The emission was detected using PMTs set at 505–530 nm. Images were processed with ImageJ and Photoshop CS6 using the same settings.

## Quantification and statistical analysis

For each experiment, numbers of ovules were collected from 10–20 plants. We used a Leica TCS SP2 laser scanning confocal microscope under Differential interference contrast (DIC) optics to count numbers of ovules with zero MMC, one MMC, and multiple MMCLs based on the MMC morphology and expression of the MMC marker *KNU::KNU-VENUS*. Meiotic MMCs were determined by callose staining. Numbers of ovules with different expression patterns of *DRSrev::GFP* were counted with a Leica TCS SP2 laser scanning confocal microscope. The "*n*" represents the total number of ovules analyzed. Three biological replicates were performed for each experiment. The statistical significance was evaluated by the Student's *t* test. For the qRT-PCR assay, three biological replicates were performed, and the data were analyzed by the comparative C(T) method[67]. Bars indicate standard deviation. Asterisks indicate $p < 0.05$.

## Reporting summary

Further information on research design is available in the Nature Portfolio Reporting Summary linked to this article.

# Data availability

The data supporting the findings in this study are available from the corresponding author upon reasonable request. Source data are provided with this paper.

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

## Acknowledgements

We thank A. Cheung, W. Lukowitz, V. Walbot, D. Weijers, and R. Yadegari for critically reading the manuscript; E. Xiong and G. Zhang for preparing some experiments, T. Schuck, J. Gonnering, and P. Engevold for plant care, the *Arabidopsis* Biological Resource Center (ABRC) for *ARF10*,

*ARF16, ARF17, EMS1, MIR160a* BAC clones and cDNAs, the *SALK_090804* seed, T. Nakagawa for pGBW vectors, Y. Zhao for the *YUC1* cDNA, Q. Chen for the pHEE401E vector, R. Yadegari for pAT5G01860::n1GFP, pAT5G45980:n1GFP, pAT5G50490::n1GFP, pAT5G56200:n1GFP vectors, and D. Weijers for the pGreenII KAN SV40-3×GFP and R2D2 vectors, W. Yang for the *spl* mutant, Y. Qin for the pKNU::KNU-VENUS vector and seed, G. Tang for the STTM160/160-48 vector, and L. Colombo for *pPIN1::PIN1-GFP spl* and *pin1-5* seeds. This work was supported by the US National Science Foundation (NSF)-Israel Binational Science Foundation (BSF) research grant to D.Z. (IOS-1322796) and T.A. (2012756). D.Z. also gratefully acknowledges supports of the Shaw Scientist Award from the Greater Milwaukee Foundation, USDA National Institute of Food and Agriculture (NIFA, 2022-67013-36294), the UWM Discovery and Innovation Grant, the Bradley Catalyst Award from the UWM Research Foundation, and WiSys and UW System Applied Research Funding Programs.

## Author contributions

J.H. and D.Z. conceived ideas and designed experiments. J.H., L.Z., S.M., B.R.G., V.X., and H.A.O. performed experiments. J.H., J.F., and D.Z. analyzed data. J.H., T.A., J.F., and D.Z. wrote the manuscript with contributions from other authors.

## Competing interests

The authors declare no competing interests.
