## [Peer Review File · Nature Communications]

Specification of female germline by microRNA orchestrated auxin signaling in ArabidopsisReviewers' Comments:

Reviewer #1:

Remarks to the Author:

Review for Huang et al "Specification of plant germline by microRNA orchestrated auxin signalling"

This manuscript addresses an interesting topic. The role of auxin during female germline development has been considered in a number of studies, but many details have been lacking. In particular, the role of the putative auxin maximum at the tip of the ovule has proved a fascinating distraction. Does it influence MMC specification or expansion? Does auxin actually enter the MMC? Is the DR5 marker truly representing endogenous auxin accumulation in the ovule? What is the relationship between auxin and ovule ARFs?

This manuscript adds some interesting components to this model, particularly in terms of ARF17 function in the ovule and its link to SPOROCTELESS. Unfortunately, the manuscript misses the mark through a lack of precision, missing molecular data and over-interpretation of results. I see a number of major issues that will prevent publication in its current form.

Comments

Page 7 line 14 to page 8 line 2: The authors suggest that miR160 accumulates in the MMC, unlike the miR160 precursor that accumulates in the chalaza. They also suggest that the lack of miR160 in foc mutants leads to the formation of multiple MMCs, and the subsequent accumulation of ARF17 in these cells. The link between these two events is unclear – why would accumulation of ARF17 in the MMC lead to the formation of another adjoining MMC?

This cell-type specific function of miR160 is potentially a very interesting finding but one that appears weakly supported since it is based on qualitative in situ hybridisation data. It is not clear where miR160 acts. Most studies of this type express a miR-sensitive GFP reporter under the control of a ubiquitous promoter (i.e. pUBI:mir160-GFP). Lack of GFP signal indicates where miR160 functions. Conversely, a miR sponge should be expressed under the control of a specific promoter to sequester functional miR160 and provide additional support the case for miRNA functioning there. In this case it would be pKNU-miR160sponge. These are standard experiments for miRNA-target studies. The lack of evidence means that the conclusion on page 8, line 1 is not well supported.

Page 8 line 8: The overexpression of mARF17 in the spl mutant is reported to restore MMC formation. This is based on observations from ovule clearing which is insufficient. Re-establishment of MMC formation needs to be supported by marker expression (i.e. pKNU:KNU-VENUS). Moreover The formation of a FM needs to be supported by a gametophyte marker such as FM1, FM2 or LC2. It is also strange that the number of ovules recovering an "MMC" (~44%) does not equal the number of ovules containing an FM (~27%). In general the quality of DIC panels in Fig 3 is poor and it is difficult to ascertain cell identity.

Page 8, Line 16: In the genetic analysis section, the authors went to some trouble to generate double mutants between weak spl and arf17 alleles. This clearly shows some enhancement of the spl phenotype. However, the where is the spl foc double mutant data. Based on the model, wouldn't an increase in ARF17 level (in foc) rescue MMC initiation in spl? Or is it really dependent upon the presence of an MMC precursor?

Page 9 line 8: Once again, the authors draw conclusions about "MMC" identity without using an appropriate marker. In this case, the pin1-5 mutants appears to show 35% extra enlarged cells in the ovule. In the absence of a marker it is not clear whether these represent MMC-like cells.

Page 9, line 10: The authors should note that similar NPA results have been published in other

species. For example, previous studies investigated the impact of NPA treatment on DR5 accumulation and formation of MMC-like cells during ovule development in *Hieracium*. Although the Koltunow lab used a different model species, they showed that NPA treatment led to an increase in DR5 signal and extra MMC-like cells (called aposporous initials; Tucker et al., 2012; JXB).

Page 9, line 14: The formation of the "extra embryo sacs" after NPA treatment looks really curious, but once again, it is not supported by expression of any marker genes. This makes it very difficult to make claims about cell identity – it needs to be carried out in a line expressing an FG marker (as per above).

Page 9, line 16: The authors use pEMS1:YUC1 in an attempt to increase the amount of auxin biosynthesis in the ovule. They highlight strong effects on ovule development including ectopic pKNU:KNU-VENUS expression. This provides hints that auxin might contribute to MMC formation. However, they don't repeat this experiment in a DR5 background to confirm that it relates to ectopic accumulation of auxin. Moreover, the majority of recent studies investigating auxin accumulation during development use the DII-VENUS line to provide additional support for the DR5 results. This marker is lacking here.

Page 11, Line 19: This statement is not correct. miR160 and ARF17 impact the number of cells expressing PIN1, not the localisation. This number of cells expressing PIN1 may relate to other factors that activate PIN1, or to direct activation via ARF17.

Page 12, line 3: The authors should indicate that the examination of PIN1:GFP in *spl* mutants has already been published (Bencievenga et al., 2012, *Plant Cell*)

Page 13, line 17: The interaction between ARF17 and SPL has been considered in this study via genetic analysis, but I am uncomfortable with statement that "ARF17 and SPL/NZZ mutually regulate each other". I believe this is derived only from qPCR data in the last paragraph of the results. Although the authors did well to collect and examine young ovules, there is no direct evidence to support this involves "regulation" rather than changes in tissue identity that impact mRNA abundance.

Page 15, line 4: The concept of one cell hypodermal cell accessing the auxin maximum and pushing other cells aside is really interesting. However, is there real evidence for this? And is it relevant for MMC formation? A number of the "multi-MMC" mutants show extra cells forming, but often these are closer to the chalaza and appear quite distal to the DR5 expressing cells.

Page 15: It is not clear to me why the increase in ARF17 leads to the multi-MMC phenotype. Is it due to the higher levels of ARF17 in the chalaza and funiculus, or to the increase in ARF17 in the MMC? I think the authors should have tested this via pKNU:mARF17 or pTAA1:mARF17.

Page 7 Line 4: The number of ovules showing extra embryo sacs in pARF17:mARF17 foc ovules is low, but only 96 were examined. This is not enough to draw any conclusions.

Reviewer #2:

Remarks to the Author:

The control of cell differentiation during female reproductive development in *Arabidopsis* is strictly controlled so that a single Megaspore Mother Cell (i.e. a cell that will enter meiosis) develops per ovule. The authors previously showed that the recessive *Arabidopsis* mutant floral organs in carpels (foc), with a transposon insertion in the 3' regulatory region of MIR160a, fails to accumulate mature miR160 in inflorescences (Liu et al., 2010, *Plant Journal*). The mutant has defects in embryogenesis. ARF10, ARF16 and ARF17 expression patterns are altered in the mutant. In this paper the authors examined this mutant and pARF17::microRNA resistant ARF17 line (and several other lines) during

female reproductive development, specifically the period before, during and after MMC formation.

Summary of figures:

Figure 1) MMC counting was performed in wild type, *foc*, *pARF17::microRNA* resistant *ARF17* and the double "mutant" line (*foc pARF17::microRNA* resistant *ARF17*) with and without presence of *pKNU::KNU-Venus* marker. Conclusion is that lack of *miRNA160* leads to overexpression of *ARF17* which causes increased "KNU positive cell" number and infertility. Whether the increased "KNU positive cell" number causes reduced fertility is not clear.

Figure 2) The authors performed IF experiments on:

pARF17::ARF17-GFP in wild type (only found signal in whole-mount IF)
pARF17::ARF17-GFP in *foc* (signal via confocal in 2 MMC)
pARF17::miRNA resistant *ARF17-GFP* in wild type (signal via confocal in 2 MMC)
pARF17::miRNA resistant *ARF17-GFP* in *foc* (signal via confocal in 2 MMC)

Authors conclude that *ARF17* protein level is much higher in *foc*, *ARF17* *miRNA* resistant line and *ARF17* *miRNA* resistant line in *foc* background

Figure 3) Authors work suggests that the classic (highly infertile) *sportocyteless* mutant can be partially suppressed by the *ARF17* *miRNA* resistant line. This is an important finding and should be robustly supported. See major comment 2.

Figure 4) *Pin1-5* mutant has more MMCs. Chemical inhibition of PIN auxin transporter (by NPA) has same effect (light microscopy and more "KNU positive cell"). Multiple MMCs enter meiosis in NPA treatment. Expression of auxin biosynthesis gene *YUC1* (*pEMS::YUC1*) led to extra meiotic MMCs.

Figure 5) Auxin response marker (*DR5rev::GFP*) present at apex of ovule primordium and nucellus in wild type. NPA treatment changes the auxin maxima in terms of position and number (stage 2-III).

Figure 6) Auxin response marker (*DR5rev::GFP*) in *foc1* mutant, *ARF17* *miRNA* resistant line and *ARF17* *miRNA* resistant line in *foc* background. *PIN1::PIN1-GFP* in *foc1* mutant, *ARF17* *miRNA* resistant line and *ARF17* *miRNA* resistant line in *foc* background

Figure 7) Auxin response marker (*DR5rev::GFP*) in *spl* mutant, Auxin response marker (*DR5rev::GFP*) in *spl* *ARF17* *miRNA* resistant line "double mutant", *PIN1::PIN1-GFP* in the same two lines.

This is very nice article that I enjoyed reading. The authors make a series of advances on a difficult topic. This paper shows that *miRNA160*, *ARF17* and auxin play an important role in regulating MMC specification in Arabidopsis. The experiments have been performed with sufficient controls and the data appears to be robust. The title is slightly too general and it should be altered to include the words "female" and "Arabidopsis". The abstract is appropriate.

I do, however, think that a series of points should be addressed to add orthogonal proof of several of the main conclusions. The paper is highly reliant on light microscopy and IF experiments (which are admittedly nice experiments) and the addition of some conventional crossing experiments would provide important orthogonal evidence for the claims that are made.

Major points

1) Figure 1q) if the second MMCs present in *foc* and *pARF17::mARF17* do not enter meiosis, perhaps it

would be better to refer to them as “MMC-like cells”, as has been used in the past in this field (e.g. Zhao et al., 2018, PNAS). The cells do not enter meiosis and therefore they may have some characteristics of MMC cells (i.e. KNU expression) but they do not enter meiosis which, for me, is a defining character of an MMC.

2) The replicate numbers in Figures 1 and 3, 4, 5, 6 and 7 clearly demonstrate the robustness of the data (i.e. the presentation of n numbers on the micrographs and in the text). No n values are presented for Figure 2. How many replicates were used? I can understand that likely less replicates were carried out for IF but please present this data. I believe this is now a standard requirement for Nature journals.

3) The authors claim “pARF17::mARF17 partially rescued the formation of MMC in the spl mutant” and “pARF17::mARF17 restored the FM formation to normal in 27% of spl ovules”. Did the authors look at female gametogenesis in this line? An important test to genetically prove the cells observed are FMs, would be by pollination using wild type pollen (i.e. wild type pollen used to pollinate spl and spl pARF17::mARF17). If MMC and FM can be truly formed in this double “mutant” background one would expect female gametophyte development and increased female fertility. If this is the case pollination of spl pARF17::mARF17 should lead to viable seeds. This will provide formal genetic proof that the cells observed are functional megaspores and can lead to the formation of female gametes.

4) The section on the CRISPR/Cas9 mutagenesis is not very clear (Figure 3 and Supp. figure 3). Which mutant (1,4 or 6?) and which generation of plants were used for phenotypic analysis in Figure 3? Also were non-transgenic segregants used or is Cas9 still present in the plants analysed. Please make this reporting clearer.

5) On page 8 line 19 and in supp figure 3 the authors show the carf17 mutant (not sure which mutant exactly?) line is male sterile. This is used to explain why the silique length is short in carf17 mutants (supp figure 3g). Yet the authors also claim FM formation is lower in a carf17 mutants. Again conventional pollinations would be insightful here. Perform controlled crosses (wild type x wild type & carf17 x wild type) side-by-side. Seed set should be lower in the carf17 x wild type cross if functional megaspore production is truly lower.

6) The authors very nicely show that mature miRNA160 accumulates in a single hypodermal cell (fig 2b). The juxtaposition of this cell just below the auxin maxima (5a) is striking. The authors examine expression of miR160 by a GFP fusion but the pattern of immature miRNA160 (2a) is not the same as mature miR160 (2b) (i.e. only the mature miRNA accumulates in the hypodermal cell). Can the authors test how mature miRNA160 accumulates when NPA treatment is used?

Minor points

Page 3 - Line 10-12 rephrase the sentence as it is not clear

Page 4 – Line 8 it would be appropriate to add citation of some reviews about small RNAs and reproduction (for instance a general one - Borges et al., 2015, Nature Reviews Mol Cell Bio - and a more ovule specific one – Petrella et al., 2021, Plant Reproduction).

Page 7 - Line 19 Please mention here you refer to the bottom left panel. I assume this is the experiment directly with confocal? It is not clear.

Page 8 – Line 8 change to “overexpressed”

Page 8 – Line 18 – Please mention the spl-3 mutant is a T-DNA allele here. It can be misunderstood to be a CRISPR mutant.

Page 9 – Line 2 – Not clear what is the conclusion here. I think it is that ARF17 is genetically downstream of SPL/NZZ?

Page 13 – Line 14-15 Please also refer to raw data figure panel (i.e. 2b for miR160 and 5a for auxin maxima) and not just model. The other parts of the discussion would also be strengthened by referring to specific figure panels (it will help the reader to find back some details when reading the discussion).

Point-by-point Responses

We thank two reviewers very much for their comments and constructive suggestions. In this resubmission, we have addressed all the concerns and revised our manuscript accordingly. Please notice that our responses and the revised texts in the manuscript are highlighted in blue.

1. Responses to Comments from Reviewer 1:

Comment:

Review for Huang et al “Specification of plant germline by microRNA orchestrated auxin signalling”

This manuscript addresses an interesting topic. The role of auxin during female germline development has been considered in a number of studies, but many details have been lacking. In particular, the role of the putative auxin maximum at the tip of the ovule has proved a fascinating distraction. Does it influence MMC specification or expansion? Does auxin actually enter the MMC? Is the DR5 marker truly representing endogenous auxin accumulation in the ovule? What is the relationship between auxin and ovule ARFs?

This manuscript adds some interesting components to this model, particularly in terms of ARF17 function in the ovule and its link to SPOROCTELESS. Unfortunately, the manuscript misses the mark through a lack of precision, missing molecular data and over-interpretation of results. I see a number of major issues that will prevent publication in its current form.

Response:

Thank you so much for your valuable comments. Our previously presented and new results suggest that auxin signaling orchestrated by the miR160-regulated *ARF17* controls the MMC specification rather than its proliferation. Using the R2D2 auxin reporter, we found that auxin entered the MMC (new Fig. 5e-h and Fig. 6g, h). The *DR5* marker does not completely represent the endogenous auxin accumulation in the ovule; however, there is an overlap at the apex of nucellus. miR160 targets *ARF10*, *ARF16* and *ARF17*, but neither *ARF10* nor *ARF16* was detected in the MMC; moreover, *mARF10* and *mARF16* are not involved in the MMC differentiation. Regarding other ARFs associated with ovule development, our results show that *ARF5* is present at epidermis of the proximal end of nucellus and the chalaza (data not shown in this manuscript). *ARF3* was found at the chalaza (Su et al., 2017). So far, we only found that *ARF17* is localized in the MMC. Auxin might directly or indirectly activate the function of *ARF17* in the MMC.

In this resubmission, our new results and revisions have addressed your concerns about missing “the mark through a lack of precision, missing molecular data and over-interpretation of results.” Please see more explanations below.

Comment:

Page 7 line 14 to page 8 line 2: The authors suggest that miR160 accumulates in the MMC, unlike the miR160 precursor that accumulates in the chalaza. They also suggest that the lack of miR160 in foc mutants leads to the formation of multiple MMCs, and the subsequent accumulation of *ARF17* in these cells. The link between these two events is unclear – why would accumulation of *ARF17* in the MMC lead to the formation of another adjoining MMC?

Response:

We appreciate your question. Our results support that the formation of extra MMCs is caused by ectopic expression of *ARF17* in nucellus hypodermal cells. To test if overexpression of *ARF17* specifically in the MMCP and MMC leads to formation of extra MMCs, we generated *pKNU::mARF17* to overexpress *ARF17* in MMCP and MMC using the MMCP and MMC specific promoter *KNU* (new Fig. 2q, r, u, v). For the same purpose, we generated *pKNU::STTM160/160-48* to specifically knock down the mature miR160 in MMCP and MMC via the STTM (Short Tandem Target Mimic) approach (new Fig. 2s, t, w, x). We observed only one MMC in ~95% of *pKNU::mARF17* and *pKNU::STTM160/160-48* ovules. Interestingly, ~5% of *pKNU::mARF17* and *pKNU::STTM160/160-48* ovules produced two fully differentiated MMCs (meiosis occurred in these MMCs). In 4.8% of WT ovules at the stage 2-I, *pKNU::KNU-VENUS* is expressed in two cells and the GFP signal in one cell is weaker than that in the other; however, after stage 2-I, the GFP signal was only observed in one MMC (Supplementary Fig. 1). Our results suggest that overexpression of *ARF17* specifically in the MMCP and MMC does not cause the formation of extra MMCs. When ovules sporadically form two MMCPs, the enhanced *ARF17* expression by *pKNU::mARF17* and *pKNU::STTM160/160-48* promotes the complete differentiation of two MMCs.

Overall, our gain- and loss-of-function studies provide evidence that *ARF17* is a key determinant for promoting the MMC specification instead of the MMC proliferation. The formation of supernumerary MMCs is caused by ectopic expression of *ARF17* in *foc*, *pARF17::mARF17*, *pARF17::mARF17 foc*, *pEMS1::YUC1*, *pin1-5*, and NPA-treated ovules. In addition, auxin signaling affects the *ARF17* function and *ARF17* in turn modulates auxin signaling via affecting the PIN1 expression domain. Please see more explanations in the Discussion section.

Comment:

This cell-type specific function of miR160 is potentially a very interesting finding but one that appears weakly supported since it is based on qualitative in situ hybridisation data. It is not clear where miR160 acts. Most studies of this type express a miR-sensitive GFP reporter under the control of a ubiquitous promoter (i.e. pUBI:mir160-GFP). Lack of GFP signal indicates where miR160 functions. Conversely, a miR sponge should be expressed under the control of a specific promoter to sequester functional miR160 and provide additional support the case for miRNA functioning there. In this case it would be pKNU-miR160sponge. These are standard experiments for miRNA-target studies. The lack of evidence means that the conclusion on page 8, line 1 is not well supported.

Response:

We are grateful for your valuable comments and suggestions. Using the *UBI10* promoter, we generated the miR160 GFP sensor *pUBI10::miR160sensor-NSL-3xGFP* (the control is *pUBI10::NSL-3xGFP*). Our results show that the mature miR160 functions in the MMC (new Fig. 2i-l), and the NPA treatment reduced accumulation of the mature miR160 in the MMC and other cells in nucellus and chalaza (new Fig. 7j-l).

Using the MMCP and MMC specific promoter *KNU*, we generated *pKNU::STTM160/160-48* to specifically knock down the mature miR160 in MMCP and MMC via the STTM approach (new Fig. 2s, t, w, x). *pKNU::STTM160/160-48* ovules mainly produced one MMC, although ~5% of ovules formed two MMCs. We obtained similar results from

analyzing *pKNU::mARF17* ovules in which *ARF17* is overexpressed in MMCP and MMC. Our results suggest that restriction of *ARF17* to the MMC by miR160 is essential for its specification and overexpression of *ARF17* in the MMC does not promote its proliferation. Please see more explanations above.

Comment:

Page 8 line 8: The overexpression of *mARF17* in the *spl* mutant is reported to restore MMC formation. This is based on observations from ovule clearing which is insufficient. Re-establishment of MMC formation needs to be supported by marker expression (i.e. *pKNU:KNU-VENUS*). Moreover The formation of a FM needs to be supported by a gametophyte marker such as FM1, FM2 or LC2. It is also strange that the number of ovules recovering an “MMC” (~44%) does not equal the number of ovules containing an FM (~27%). In general the quality of DIC panels in Fig 3 is poor and it is difficult to ascertain cell identity.

Response:

Thank you for your comments. As the callose deposition occurs during meiosis in MMC; thus, callose staining is used to determine identity of the MMC (Mendes, M. A. *et al.*, 2020, Development; Olmedo-Monfil, V. *et al.* 2010, Nature; Su *et al.*, 2020, Plant Cell). The callose deposition indicates the completion of MMC differentiation. It is time consuming and technically challenging for us to introduce *pKNU:KNU-VENUS* into *pARF17:mARF17 spl* background; therefore, we confirmed the MMC fate restoration by callose staining in *pARF17:mARF17 spl* ovules (new Fig. 3g, k).

To examine whether restored MMCs can lead to formation of truly functional FMs, we used wild-type pollen to pollinate *spl* and *pARF17:mARF17 spl*. We observed ~17% of developing seeds (7.4/43.1; 7.4±3.7 per silique) comparing with the wild-type control (new Supplementary Fig. 3a-c), suggesting that overexpression of *ARF17* can rescue the female gametogenesis in the *spl* mutant background.

In *pARF17:mARF17 spl* plants, we observed ~44% of ovules with recovered MMCs and meiosis occurred in ~30% of MMCs (Fig. 3c and Fig. 3g); however, ~27% of ovules contained FM (Fig. 3o) and seed production rate was ~17% (new Supplementary Fig. 3a-c). Thus, our results suggest that some of MMCs did not enter meiosis and ~63% of observed FMs (17%/27%) developed into functional embryo sacs.

Comment:

Page 8, Line 16: In the genetic analysis section, the authors went to some trouble to generate double mutants between weak *spl* and *arf17* alleles. This clearly shows some enhancement of the *spl* phenotype. However, where is the *spl foc* double mutant data. Based on the model, wouldn't an increase in *ARF17* level (in *foc*) rescue MMC initiation in *spl*? Or is it really dependent upon the presence of an MMC precursor?

Response:

We appreciate your thoughts. We generated the *spl foc* double mutant. Similar to *pARF17:mARF17*, the *foc* mutation can also rescue formation of MMC, FM, and seeds in the *spl* mutant background (new Fig. 3d, h, i, p and new Supplementary Fig. 3a, b, d).

Comment:

Page 9 line 8: Once again, the authors draw conclusions about “MMC” identity without using an appropriate marker. In this case, the *pin1-5* mutants appears to show 35% extra enlarged cells in the ovule. In the absence of a marker it is not clear whether these represent MMC-like cells.

Response:

We introduced the *pKNU:KNU-VENUS* marker into the *pin1-5* mutant and confirmed the identity of extra MMCs by this marker (new Fig. 4c).

Comment:

Page 9, line 10: The authors should note that similar NPA results have been published in other species. For example, previous studies investigated the impact of NPA treatment on DR5 accumulation and formation of MMC-like cells during ovule development in *Hieracium*. Although the Koltunow lab used a different model species, they showed that NPA treatment led to an increase in DR5 signal and extra MMC-like cells (called aposporous initials; Tucker et al., 2012; JXB).

Response:

We appreciate your information. We discussed the findings by Tucker et al. in the Discussion section: “In the apomictic *Hieracium* subgenus *Pilosella* ovule, NPA treatment increases the expression level of *DR5::GFP* and alters the *DR5::GFP* expression domain. However, NPA treatment does not affect the MMC differentiation, although the number of aposporous initial cells is increased, suggesting that auxin signaling might play a more dominant role in the apomixis process in apomictic species.”

Comment:

Page 9, line 14: The formation of the “extra embryo sacs” after NPA treatment looks really curious, but once again, it is not supported by expression of any marker genes. This makes it very difficult to make claims about cell identity – it needs to be carried out in a line expressing an FG marker (as per above).

Response:

We confirmed the development of female gametophyte after NPA treatment using *pAT5G01860::n1GFP* as the FG marker (Supplementary Fig. 5q-x).

Comment:

Page 9, line 16: The authors use *pEMS1:YUC1* in an attempt to increase the amount of auxin biosynthesis in the ovule. They highlight strong effects on ovule development including ectopic *pKNU:KNU-VENUS* expression. This provides hints that auxin might contribute to MMC formation. However, they don’t repeat this experiment in a DR5 background to confirm that it relates to ectopic accumulation of auxin. Moreover, the majority of recent studies investigating auxin accumulation during development use the *DII-VENUS* line to provide additional support for the DR5 results. This marker is lacking here.

Response:

We appreciate your valuable comments and suggestions. We generated *pEMS1:YUC1 DR5::GFP* and found similar ectopic expression changes of *DR5::GFP* to that of NPA treated

WT, *DR5::GFP foc*, *DR5::GFP pARF17::mARF17*, and *DR5::GFP pARF17::mARF17 foc* ovules (new Supplementary Fig. 7).

As suggested, we examined accumulation of auxin in ovules using the R2D2 auxin reporter (*RPS5A::DII-VENUS/RPS5A::mDII- ntdTomato*). We found the accumulation of auxin in nucellus including the MMC (new Fig. 5e, f). NPA treatment and the *foc* mutation cause more accumulation of auxin in the nucellus cells and in the chalaza (new Fig. 5g, h and new Fig. 6g, h). Our results show that auxin enters MMC; the *DR5* marker does not completely represent the endogenous auxin accumulation in the ovule; however, there is an overlap at the apex of nucellus.

Comment:

Page 11, Line 19: This statement is not correct. miR160 and ARF17 impact the number of cells expressing PIN1, not the localisation. This number of cells expressing PIN1 may relate to other factors that activate PIN1, or to direct activation via ARF17.

Response:

We agree with your correction and thoughts, thank you. In this resubmission, we changed the sentence “Our results suggest that miR160 and *ARF17* affect the localization of PIN1, which is critical for establishment of a single local auxin maximum at the ovule apex.” to “Our results suggest that miR160 and *ARF17* affect expression domains of PIN1, which is critical for establishment of a single local auxin maximum at the ovule apex and accumulation of auxin in nucellus, including the MMC.”

Comment:

Page 12, line 3: The authors should indicate that the examination of PIN1:GFP in *spl* mutants has already been published (Bencievenga et al., 2012, Plant Cell)

Response:

We cited the work from Bencivenga et al., 2012, Plant Cell. Thank you.

Comment:

Page 13, line 17: The interaction between ARF17 and SPL has been considered in this study via genetic analysis, but I am uncomfortable with statement that “ARF17 and SPL/NZZ mutually regulate each other”. I believe this is derived only from qPCR data in the last paragraph of the results. Although the authors did well to collect and examine young ovules, there is no direct evidence to support this involves “regulation” rather than changes in tissue identity that impact mRNA abundance.

Response:

We agree with you and are thankful for your comments. We changed “Moreover, *ARF17* and *SPL/NZZ* mutually regulates each other.” into “Moreover, *ARF17* genetically interacts with *SPL/NZZ* (Fig. 3), and they possibly affect each other’s expression (Fig. 7q).”

Comment:

Page 15, line 4: The concept of one cell hypodermal cell accessing the auxin maximum and pushing other cells aside is really interesting. However, is there real evidence for this? And is it

relevant for MMC formation? A number of the “multi-MMC” mutants show extra cells forming, but often these are closer to the chalaza and appear quite distal to the DR5 expressing cells.

Response:

We agree with you and are thankful for your comments. We removed the sentence of “One hypodermal cell that has the greatest contact area with the auxin maximum enlarges and pushes other hypodermal cells away from the auxin maximum.” because this idea is speculative.

Comment:

Page 15: It is not clear to me why the increase in ARF17 leads to the multi-MMC phenotype. Is it due to the higher levels of ARF17 in the chalaza and funiculus, or to the increase in ARF17 in the MMC? I think the authors should have tested this via *pKNU:mARF17* or *pTAA1:mARF17*.

Response:

Thank you very much for your comments. As suggested, we generated *pKNU::mARF17* to overexpress ARF17 specifically in MMCP and MMC using the MMCP and MMC specific promoter *KNU*. As explained above, overexpression of *ARF17* in MMCP and MMC does not promote MMC proliferation. Our results from gain- and loss-of-function studies support that ARF17 specifies the MMC fate. ARF17 also modulates auxin signaling via affecting the PIN1 expression domain. When a nucellus hypodermal cell other than the MMCP expresses sufficient *ARF17* and accumulates sufficient auxin, it will acquire the MMC identity. The ARF17 is ectopically expressed and auxin signaling is abnormal in *foc*, *pARF17::mARF17*, *pARF17::mARF17 foc*, *pEMS1::YUC1*, *pin1-5*, and NPA-treated ovules, which explains why multiple MMCs are formed in these ovules.

Comment:

Page 7 Line 4: The number of ovules showing extra embryo sacs in *pARF17:mARF17 foc* ovules is low, but only 96 were examined. This is not enough to draw any conclusions.

Response:

We examined 120 more *pARF17::mARF17 foc* ovules (totally 216) and found that a similar percentage of (from 5.2%, n = 96 to 5.6%, n = 216) ovules contains two embryo sacs (Supplementary Fig. 2t, 5.6%, n = 216).

2. Responses to Comments from Reviewer 2:

Comment:

The control of cell differentiation during female reproductive development in Arabidopsis is strictly controlled so that a single Megaspore Mother Cell (i.e. a cell that will enter meiosis) develops per ovule. The authors previously showed that the recessive Arabidopsis mutant floral organs in carpels (*foc*), with a transposon insertion in the 3' regulatory region of MIR160a, fails to accumulate mature miR160 in inflorescences (Liu et al., 2010, Plant Journal). The mutant has defects in embryogenesis. ARF10, ARF16 and ARF17 expression patterns are altered in the mutant. In this paper the authors examined this mutant and *pARF17::microRNA* resistant ARF17 line (and several other lines) during female reproductive development, specifically the period

before, during and after MMC formation.

Summary of figures:

Figure 1) MMC counting was performed in wild type, *foc*, *pARF17::miRNA* resistant *ARF17* and the double “mutant” line (*foc pARF17::miRNA* resistant *ARF17*) with and without presence of *pKNU::KNU-Venus* marker. Conclusion is that lack of *miRNA160* leads to overexpression of *ARF17* which causes increased “*KNU* positive cell” number and infertility. Whether the increased “*KNU* positive cell” number causes reduced fertility is not clear.

Figure 2) The authors performed IF experiments on:

pARF17::ARF17-GFP in wild type (only found signal in whole-mount IF)
pARF17::ARF17-GFP in *foc* (signal via confocal in 2 MMC)
pARF17::miRNA resistant *ARF17-GFP* in wild type (signal via confocal in 2 MMC)
pARF17::miRNA resistant *ARF17-GFP* in *foc* (signal via confocal in 2 MMC)

Authors conclude that *ARF17* protein level is much higher in *foc*, *ARF17 miRNA* resistant line and *ARF17 miRNA* resistant line in *foc* background

Figure 3) Authors work suggests that the classic (highly infertile) *sportocyteless* mutant can be partially suppressed by the *ARF17 miRNA* resistant line. This is an important finding and should be robustly supported. See major comment 2.

Figure 4) *Pin1-5* mutant has more MMCs. Chemical inhibition of PIN auxin transporter (by NPA) has same effect (light microscopy and more “*KNU* positive cell”). Multiple MMCs enter meiosis in NPA treatment. Expression of auxin biosynthesis gene *YUC1* (*pEMS::YUC1*) led to extra meiotic MMCs.

Figure 5) Auxin response marker (*DR5rev::GFP*) present at apex of ovule primordium and nucellus in wild type. NPA treatment changes the auxin maxima in terms of position and number (stage 2-III).

Figure 6) Auxin response marker (*DR5rev::GFP*) in *foc1* mutant, *ARF17 miRNA* resistant line and *ARF17 miRNA* resistant line in *foc* background. *PIN1::PIN1-GFP* in *foc1* mutant, *ARF17 miRNA* resistant line and *ARF17 miRNA* resistant line in *foc* background

Figure 7) Auxin response marker (*DR5rev::GFP*) in *spl* mutant, Auxin response marker (*DR5rev::GFP*) in *spl ARF17 miRNA* resistant line “double mutant”, *PIN1::PIN1-GFP* in the same two lines.

This is very nice article that I enjoyed reading. The authors make a series of advances on a difficult topic. This paper shows that *miRNA160*, *ARF17* and auxin play an important role in regulating MMC specification in *Arabidopsis*. The experiments have been performed with sufficient controls and the data appears to be robust. The title is slightly too general and it should be altered to include the words “female” and “*Arabidopsis*”. The abstract is appropriate.

I do, however, think that a series of points should be addressed to add orthogonal proof of several of the main conclusions. The paper is highly reliant on light microscopy and IF experiments (which are admittedly nice experiments) and the addition of some conventional crossing experiments would provide important orthogonal evidence for the claims that are made.

Response:

We are so thankful for your valuable comments and suggestions. We have changed the title “Specification of plant female germline by microRNA orchestrated auxin signaling” to “Specification of female germline by microRNA orchestrated auxin signaling in *Arabidopsis*”.

As suggested, we used wild-type pollen to pollinate *spl*, *pARF17:mARF17 spl*, and *spl foc* plants. Our results show that overexpression of *ARF17* can rescue the female gametogenesis in the *spl* mutant background (new Supplementary Fig. 3a-d). Using the same approach, we also show that the *carf17* mutant is partially defective in female gametogenesis (new Supplementary Fig. 3a, e).

Our results showed that increased “KNU positive cell” numbers reduced fertility of *foc*, *pARF17::mARF17*, and *pARF17::mARF17 foc* plants (Please see Supplementary Table 1).

In this resubmission, we have addressed all your concerns. Please see detailed explanations below.

Comment:

Major points

1) Figure 1q) if the second MMCs present in *foc* and *pARF17::mARF17* do not enter meiosis, perhaps it would be better to refer to them as “MMC-like cells”, as has been used in the past in this field (e.g. Zhao et al., 2018, PNAS). The cells do not enter meiosis and therefore they may have some characteristics of MMC cells (i.e. KNU expression) but they do not enter meiosis which, for me, is a defining character of an MMC.

Response:

Thank you so much for your suggestion. In this resubmission, we used “MMC-like (MMCL) cells” to refer to supernumerary MMCs expressing the KNU marker and identified by DIC. We still named “Extra MMCs” that enter meiosis as the MMC.

Comment:

2) The replicate numbers in Figures 1 and 3, 4, 5, 6 and 7 clearly demonstrate the robustness of the data (i.e. the presentation of n numbers on the micrographs and in the text). No n values are presented for Figure 2. How many replicates were used? I can understand that likely less replicates were carried out for IF but please present this data. I believe this is now a standard requirement for Nature journals.

Response:

We appreciate your comment and suggestion. We added numbers of ovules to the Figure 2. We examined many ovules because we performed whole-mount RNA *in situ* hybridization and whole-mount immunolocalization for analyzing expressions of *MIR160a*, miR160, and *ARF17* at transcriptional and translational levels.

Comment:

3) The authors claim “pARF17::mARF17 partially rescued the formation of MMC in the *spl* mutant” and “pARF17::mARF17 restored the FM formation to normal in 27% of *spl* ovules”. Did the authors look at female gametogenesis in this line? An important test to genetically prove the cells observed are FMs, would be by pollination using wild type pollen (i.e. wild type pollen used to pollinate *spl* and *spl* pARF17::mARF17). If MMC and FM can be truly formed in this double “mutant” background one would expect female gametophyte development and increased female fertility. If this is the case pollination of *spl* pARF17::mARF17 should lead to viable seeds. This will provide formal genetic proof that the cells observed are functional megaspores and can lead to the formation of female gametes.

Response:

We are so thankful for your suggestions. We used wild-type pollen to pollinate *spl* and *pARF17:mARF17 spl*. We found that ~17% (7.4/43.1; 7.4 ± 3.7 per silique) of developing seeds was rescued comparing with the wild-type control (new Supplementary Fig. 3a-c). Our results suggest that overexpression of ARF17 could rescue the female gametogenesis in the *spl* mutant background.

In *pARF17:mARF17 spl* plants, ~44% of ovules contain the rescued MMC and meiosis occurred in ~30% of MMCs (Fig. 3c and Fig. 3g); however, FM was found in ~27% of ovules (Fig. 3o) and seed production rate was ~17% (new Supplementary Fig. 3a-c). Therefore, our results suggest that some of MMCs did not enter meiosis and ~63% of observed FMs (17%/27%) developed into functional embryo sacs, and finally seeds.

Comment:

4) The section on the CRISPR/Cas9 mutagenesis is not very clear (Figure 3 and Supp. figure 3). Which mutant (1,4 or 6?) and which generation of plants were used for phenotypic analysis in Figure 3? Also were non-transgenic segregants used or is Cas9 still present in the plants analysed. Please make this reporting clearer.

Response:

We appreciate your questions. Among three *carf17* mutants that we generated, the *carf17-6* which has no Cas9 was used for detailed analysis. For simplicity, we still used the name *carf17* in this paper. We have added this information to our revised manuscript.

Comment:

5) On page 8 line 19 and in supp figure 3 the authors show the *carf17* mutant (not sure which mutant exactly?) line is male sterile. This is used to explain why the silique length is short in *carf17* mutants (supp figure 3g). Yet the authors also claim FM formation is lower in a *carf17* mutants. Again conventional pollinations would be insightful here. Perform controlled crosses (wild type x wild type & *carf17* x wild type) side-by-side. Seed set should be lower in the *carf17* x wild type cross if functional megaspore production is truly lower.

Response:

Thank you so much for your suggestion. As explained above, we generated three *carf17* independent mutants. The *carf17-6* was used for detailed analysis. We used wild-type pollen to pollinate *carf17-6* and wild-type plants. We observed 23.2% (1-33.1/43.1; 33.1 ± 4.4 per silique) of seed reduction in comparison with the control (new Supplementary Fig. 3a, e). We found

14.7% of failure of FM formation (Fig. 3r, n = 136). Thus, our results suggest that some FMs might not develop into embryo sacs or some embryo sacs possibly failed to develop into seeds.

Comment:

6) The authors very nicely show that mature miRNA160 accumulates in a single hypodermal cell (fig 2b). The juxtaposition of this cell just below the auxin maxima (5a) is striking. The authors examine expression of miR160 by a GFP fusion but the pattern of immature miRNA160 (2a) is not the same as mature miR160 (2b) (i.e. only the mature miRNA accumulates in the hypodermal cell). Can the authors test how mature miRNA160 accumulates when NPA treatment is used?

Response:

We appreciate your suggestion. Using the *UBI10* promoter, we generated the miR160 GFP sensor *pUBI10::miR160sensor-NSL-3xGFP*. Our results show that the mature miR160 acts in the MMC (new Fig. 2i-l). NPA treatment decreased accumulation of the mature miR160 in the MMC and other cells in nucellus and chalaza (Fig. 7j-l), which explains why NPA treatment causes supernumerary MMCLs.

Comment:

Minor points

Page 3 - Line 10-12 rephrase the sentence as it is not clear

Response:

Thank you for your comment. We have modified this sentence to “Following double fertilization, the seed which typically harbors a single sexually produced embryo is eventually formed.”

Comment:

Page 4 – Line 8 it would be appropriate to add citation of some reviews about small RNAs and reproduction (for instance a general one - Borges et al., 2015, Nature Reviews Mol Cell Bio - and a more ovule specific one – Petrella et al., 2021, Plant Reproduction).

Response:

We have cited these two papers and two more other review articles.

Comment:

Page 7 - Line 19 Please mention here you refer to the bottom left panel. I assume this is the experiment directly with confocal? It is not clear.

Response:

This has been changed to “(Fig. 2m, the bottom left inset)”. Yes, this image was acquired by a confocal microscope. Thank you.

Comment:

Page 8 – Line 8 change to “overexpressed”

Response:

This has been changed. Thank you.

Comment:

Page 8 – Line 18 – Please mention the spl-3 mutant is a T-DNA allele here. It can be misunderstood to be a CRISPR mutant.

Response:

This information has been added.

Comment:

Page 9 – Line 2 – Not clear what is the conclusion here. I think it is that ARF17 is genetically downstream of SPL/NZZ?

Response:

Thanks for your correction. We have changed “Collectively, our results suggest that *ARF17* is required for promoting MMC specification by genetically with *SPL/NZZ*.” into “Collectively, our results suggest that *ARF17* is required for promoting MMC specification by genetically acting downstream of *SPL/NZZ*.”

Comment:

Page 13 – Line 14-15 Please also refer to raw data figure panel (i.e. 2b for miR160 and 5a for auxin maxima) and not just model. The other parts of the discussion would also be strengthened by referring to specific figure panels (it will help the reader to find back some details when reading the discussion).

Response:

We are thankful for your suggestions. We added specific figure panels in the Discussion section.

Reviewers' Comments:

Reviewer #1:

Remarks to the Author:

In this revised version of the manuscript, the authors have added new data and addressed many of the concerns raised in the first round of review.

Despite my enthusiasm for the data and story, there are still a number of gaps in logic and flow that make it difficult to understand the authors conclusions. There seem to be a number of issues with precision and incorrect use of terminology in reference to MMC development i.e. proliferation vs specification. The lack of precision at certain points of the manuscript makes it difficult for the reader to interpret the results. Unfortunately this lack of clarity gives the impression that the authors are not totally sure of their conclusions and what they mean in terms of ovule development.

I have tried to identify both minor and major issues in detail below.

Line 29: Does auxin determines the MMC fate? I would suggest that auxin promotes or contributes to MMC fate.

Line 34: there is no evidence that PIN1 localization is impacted by miR160 or ARF17. The domain of PIN1 expression is expanded (see more comments below)

Line 35: "undocumented mechanisms". As the authors confirm in the revised version, previous reports from the Colombo lab suggested that PIN1 was downstream of SPL/NZZ function in MMC formation. Hence, change to "Our findings elucidate the mechanism by which auxin signalling promotes the acquisition of female germline cell fate in plants"

Line 50: "of a germline"

Line 51 "of the flower"

Lines 59-79 is possibly the hardest part of the whole manuscript to read. For example, lines 60-64 try to connect the KRPs/ICKs to RBR1 to WUS which is fine, but the incorrect use of tense, gene functions, names and identities convolutes the sentence structure. Non-specialists will not follow this section - please rewrite.

Line 64: "In rice and maize, a Leucine...."

Line 67: My understanding of the MSP/MAC1 pathway is that it promotes somatic identity in cells surrounding the MMC, not that it "impedes proliferation of MMC".

Line 68, 70: The term "MMC surrounding somatic cells" is not easy to understand. Please modify throughout the text to "Somatic cells surrounding the MMC"

Line 70: "...somatic cells surrounding the MMC, which prevents them from acquiring MMC identity"

Line 72: "...ARF3 in cells neighbouring the MMC to inhibit the formation of ectopic MMCs"

Line 77-79: "Although significant progress has been made towards understanding pathways that restrict MMC formation, the molecular mechanism underlying the promotion of MMC identity has remained elusive."

Line 84, 93, 151 and throughout the document: please use "expression" rather than "expressions" when referring to gene expression.

Line 85: "For example, during root..."

Line 94,95: "ARF17 specifies the MMC by..."

Line 97: "localization" is not the correct word here. These factors define the "expression domain" of PIN1. Localization refers to membranes, walls, organelles, nucleus...

Line 138, Fig 1h – the ovule looks very large and somewhat distended. Does the pARF17:mARF17 foc mutant combination have dramatic impacts on ovule size (particularly that of the nucellus) in general?

Line 141, Fig 1m: The aniline blue staining in ovules has not worked particularly well for some reason. The ovule in 1m appears to show a strange atypical dyad and staining is very weak. I agree that the staining in 1o suggests that there are multiple cells accumulating callose, but I am uncomfortable with the suggestion that this indicates they have "undergone meiosis". The MMC in Arabidopsis accumulates callose prior to meiotic division. Hence, it is OK to use callose as a marker for MMC identity, but without the addition of some confocal microscopy and meiotic antibody labelling, aniline blue labelling it is not sufficient to suggest that the ectopic MMCLs have entered meiosis.

Line 152, figure 2 a,b: The wholemount in situs are really nice. However, I think the authors need to be really careful with the MIR160a/mir160 "movement" conclusion on Lines 171/172. On my computer screen and printout, it appears that MIR160a is expressed in the chalaza, funiculus, hypodermal cells, and MMC. The pattern appears identical to miR160, although the staining in the MMC is somewhat darker.

Although the model for movement of miR160 into the MMC is attractive, and the pMIR160a reporter shown in 2g (pMIR160a-3xnlsgFP) is apparently not detected in the MMC, it's hard to argue with the in situ data. I don't think it is possible to discount the possibility that MIR160a is normally expressed in the MMC, processed there and buffers the levels of ARF17. This appears to be consistent with the overall in situ data which shows that both miR160 and ARF17 are present in the majority of the ovule cells (at the same time), but miR160 reduces ARF17 mRNA levels rather than turns them off. This is also consistent with the miR160-sensitive UBI-GFP reporter that shows an overall reduction in GFP expression.

Hence, Line 160/161 needs to be written carefully: "...suggesting that mature miR160 is active in a range of ovule cells, and in particular, the MMC."

Line 165, Fig 2m: It's confusing how the immunolabelling of the pARF17:ARF17-GFP reporter is presented. I suggest showing the "inset image" as the main figure to allow comparisons to 2n, and show the immunolabelling as the inset.

Line 167: At this point in the manuscript I still don't think the authors have established a role for ARF17 in primary MMC "specification". This would need to come from mutant analysis (line 185 onwards), so it seems premature to use this text here.

Line 173: The authors once again refer to "proliferation of MMC". I'm not sure whether this is a mistake or I am misunderstanding the phenotype. The majority of data points towards miR160 restricting ARF17 expression in the ovule, which prevents more than one MMC from forming. Hence, it's a "cell specification" defect in somatic cells rather than a "cell proliferation" defect of the MMC. I suggest changing the text... "To test whether expression of ARF17 and miR160 in the MMC is required for ectopic MMCL formation we first overexpressed miRNA resistant ARF17..."

Line 182-184: Once again, the authors appear to have delivered a conclusion that is not based on the data presented. So far in the manuscript there is no evidence to suggest that "restriction of ARF17 to the MMC by miR160 is essential for MMC specification?". The evidence from foc and pARF17:mARF17-

GFP suggests that miR160 restricts ARF17 expression in the ovule and that several MMCL cells start to enlarge and accumulate more ARF17.

Line 183/4: This is an important finding because it shows that the defects observed in foc (i.e. multiple MMCL formation) are not actually dependent upon expression of ARF17 or miR160 in the MMC itself. Instead, they most likely relate to other defects in the ovule (such as the modified vasculature?).

Line 186: Now we get to the experiments regarding specification!! Please correct the sentence "To test whether ARF17 is required for MMC specification...."

Line 224: "indicated that multiple MMCs acquired MMC identity."

Line 228: "observed multiple MMCs accumulating callose"

Line 219 onwards: Please define here how NPA treatments were carried out? At subsequent points in the text there are references to 1, 2 and 4 days of treatment, the significance of which is unclear.

Line 229/230: Please consider rephrasing "Hence, disruption of PIN-dependent PAT and increased local auxin biosynthesis lead to ectopic MMC formation"

Line 260-263: I don't understand this sentence. Please be precise. What is "the abnormal MMC specification"

Line 278 onwards: I actually think that the changes in vascular tissue (marked by PIN1 accumulation in a larger domain) may have a lot to do with the defects. They seem to correlate with wider ovules (Fig 6j,k,l) – is this the case? One concern is that the ovule in Figure 6J doesn't appear to show this broader domain of PIN1 expression in a convincing manner. The supplementary data Fig9b is much more convincing, but I am confused why the authors use such a bad ovule in 6j.

Line 293-298: It's unclear to me what this part of the text is meant to be describing. It's essentially a repeat of the results from the previous sections. Perhaps I'm missing something.

Line 321: "Here we report that..."

Line 322: The model is very difficult to interpret. The different coloured spots cannot be distinguished.

Line 324: Once again, please be cautious with the mobile miR160 conclusion (see above comments)

Line 309, Fig 7q. Please show the qPCR data in a more appropriate form. The Figure is dominated by the mARF17 expression and it is difficult to see what happens to the other genes.

Line 353: "does not result in"

Line 376: "a portion of them accumulate callose suggesting they are preparing to undergo meiosis"

Reviewer #2:

Remarks to the Author:

I appreciate the additions made by the authors which have strengthened the manuscript. Thank you for the efforts you have made to address my comments.

The demonstration that foc spl double mutants produce some MMC and FM is an important finding and

corroborates the earlier reported finding that the pARF17:mARF17 construct can partially suppress the spl phenotype.

More importantly for this reviewer are the elegant hybridization experiments described in supplementary figure 3. The crossing experiments are a completely orthogonal approach and clearly prove that that the pARF17:mARF17 construct and foc mutation can lead to MMC production and functional megaspores (and subsequently viable female gametophytes) in an spl mutant background. Genetics does not lie and I find this is a nice addition. I am not aware of other spl suppressors and I am surprised the authors do not make this a panel of a main figure (as it is a breakthrough in my eyes) but I accept their freedom to judge for themselves.

I also appreciate the addition of the pUBI10::miR160sensor-NSL-3xGFP data which shows that miR160 is present in the MMC.

The requested changes to the title and text have been made and improve the clarity of the paper.

Congratulations to the authors on this nice study!

Point-by-point Responses

We thank two reviewers very much for their valuable comments and constructive suggestions. In this resubmission, we have addressed all the concerns and revised our manuscript accordingly. Please notice that our responses and the revised texts in the manuscript are highlighted in blue. Please also notice the changes of line numbers due to revisions.

Reviewer #1 (Remarks to the Author):

In this revised version of the manuscript, the authors have added new data and addressed many of the concerns raised in the first round of review.

Despite my enthusiasm for the data and story, there are still a number of gaps in logic and flow that make it difficult to understand the authors conclusions. There seem to be a number of issues with precision and incorrect use of terminology in reference to MMC development i.e. proliferation vs specification. The lack of precision at certain points of the manuscript makes it difficult for the reader to interpret the results. Unfortunately this lack of clarity gives the impression that the authors are not totally sure of their conclusions and what they mean in terms of ovule development.

I have tried to identify both minor and major issues in detail below.

We are so grateful for your valuable suggestions and insightful comments. Please see the changes that we made below.

Line 29: Does auxin determines the MMC fate? I would suggest that auxin promotes or contributes to MMC fate.

New Line 28-29: We have changed “Here we report that spatially restricted auxin signaling determines the MMC fate.” into “Here we report that spatially restricted auxin signaling promotes MMC fate in *Arabidopsis*.”

Line 34: there is no evidence that PIN1 localization is impacted by miR160 or ARF17. The domain of PIN1 expression is expanded (see more comments below)

New Line 34: We have changed “localization” into “expression domain” for PIN1 throughout the manuscript.

Line 35: “undocumented mechanisms”. As the authors confirm in the revised version, previous reports from the Colombo lab suggested that PIN1 was downstream of SPL/NZZ function in MMC formation. Hence, change to “Our findings elucidate the mechanism by which auxin signalling promotes the acquisition of female germline cell fate in plants”

New Line 35: We have changed to “Our findings reveal a previously undocumented mechanism for promoting acquisition of the female germline cell fate in plants.” into “Our findings elucidate the mechanism by which auxin signaling promotes the acquisition of female germline cell fate in

plants.”

Line 50: “of a germline”

New Line 50: This has been changed.

Line 51 “of the flower”

New Line 51: This has been changed.

Lines 59-79 is possibly the hardest part of the whole manuscript to read. For example, lines 60-64 try to connect the KRPs/ICKs to RBR1 to WUS which is fine, but the incorrect use of tense, gene functions, names and identities convolutes the sentence structure. Non-specialists will not follow this section - please rewrite.

New Line 58-66: We have rewritten this part.

Line 64: “In rice and maize, a Leucine....”

New Line 66: This has been changed.

Line 67: My understanding of the MSP/MAC1 pathway is that it promotes somatic identity in cells surrounding the MMC, not that it “impedes proliferation of MMC”.

New Line 69-70: This has been changed.

Line 68, 70: The term “MMC surrounding somatic cells” is not easy to understand. Please modify throughout the text to “Somatic cells surrounding the MMC”

New Line 69-70: We have changed “MMC surrounding somatic cells” to “somatic cells surrounding the MMC” throughout the manuscript (e.g., line 72 and line 118).

Line 70: “...somatic cells surrounding the MMC, which prevents them from acquiring MMC identity”

New Line 71-72: We have changed “...MMC surrounding somatic cells, which inhibits these cells to acquire the MMC identity.” into “...somatic cells surrounding the MMC, which suppresses them to acquire the MMC identity.”

Line 72: “...ARF3 in cells neighbouring the MMC to inhibit the formation of ectopic MMCs”

New Line 73-74: We have changed “...ARF3 in MMC neighbor cells to inhibit the ectopic formation of multiple MMCs.” into “...ARF3 in cells neighboring the MMC to inhibit the formation of ectopic MMCs.”

Line 77-79: “Although significant progress has been made towards understanding pathways that

restrict MMC formation, the molecular mechanism underlying the promotion of MMC identity has remained elusive.”

New Line 78-80: We have changed “Although significant progress has been made in understanding restricting MMC formation, the molecular mechanism underlying promoting MMC specification remains mysterious.” into “Although significant progress has been made towards understanding pathways that restrict MMC formation, the molecular mechanism underlying the promotion of MMC identity remains elusive.”

Line 84, 93, 151 and throughout the document: please use “expression” rather than “expressions” when referring to gene expression.

We have changed “expressions” into “expression” throughout the manuscript.

Line 85: “For example, during root...”

New Line 86: This has been changed.

Line 94,95: “ARF17 specifies the MMC by...”

New Line 96: This has been changed.

Line 97: “localization” is not the correct word here. These factors define the “expression domain” of PIN1. Localization refers to membranes, walls, organelles, nucleus...

New Line 98: We have changed “localization” into “expression domain” for PIN1 throughout the manuscript.

Line 138, Fig 1h – the ovule looks very large and somewhat distended. Does the *pARF17:mARF17* foc mutant combination have dramatic impacts on ovule size (particularly that of the nucellus) in general?

We observed enlarged nucelli in *pARF17:mARF17 foc* ovules where more than two MMCLs were produced. The extra MMCLs might contribute to the increase of nucellus size.

Line 141, Fig 1m: The aniline blue staining in ovules has not worked particularly well for some reason. The ovule in 1m appears to show a strange atypical dyad and staining is very weak. I agree that the staining in 1o suggests that there are multiple cells accumulating callose, but I am uncomfortable with the suggestion that this indicates they have “undergone meiosis”. The MMC in *Arabidopsis* accumulates callose prior to meiotic division. Hence, it is OK to use callose as a marker for MMC identity, but without the addition of some confocal microscopy and meiotic antibody labelling, aniline blue labelling it is not sufficient to suggest that the ectopic MMCLs have entered meiosis.

New Line 139-143: We replaced the image in Fig. 1m with a new image, showing the MMC at the dyad stage (Qin et al., Plant Cell 2014). We have changed “In addition, analysis of callose

deposition which marks ongoing meiosis^{14,16}, found that meiosis typically occurred only in one MMC in WT, *loc*, and *pARF17::mARF17* ovules (Fig. 1m, n, 92.6%, n = 122 and Fig. 1q), whereas two MMCs underwent meiosis in *pARF17::mARF17 loc* ovules (Fig. 1o, p, 18.9%, n = 175 and Fig. 1q).” into “In addition, analysis of callose deposition that was used as a cytological marker for MMC ongoing meiosis^{14,16}, found that meiosis typically occurred only in one MMC in WT, *loc*, and *pARF17::mARF17* ovules (Fig. 1m, n, 92.6%, n = 122 and Fig. 1q), whereas two MMCs are preparing to enter meiosis in *pARF17::mARF17 loc* ovules (Fig. 1o, p, 18.9%, n = 175 and Fig. 1q).” Similar changes were also made in other places.

Line 152, figure 2 a,b: The wholemount in situ are really nice. However, I think the authors need to be really careful with the MIR160a/mir160 “movement” conclusion on Lines 171/172. On my computer screen and printout, it appears that MIR160a is expressed in the chalaza, funiculus, hypodermal cells, and MMC. The pattern appears identical to miR160, although the staining in the MMC is somewhat darker.

New Line 172: We have changed “Our results suggest that the mature miR160 is possibly synthesized in chalaza and funiculus, then accumulates in the MMC, where miR160 negatively regulates the expression of *ARF17*.” into “Our results suggest that the mature miR160 negatively regulates the expression of *ARF17*.” We did not state “movement” of miR160 in this revised manuscript.

Although the model for movement of miR160 into the MMC is attractive, and the pMIR160a reporter shown in 2g (pMIR160a-3xnlGFP) is apparently not detected in the MMC, it’s hard to argue with the in situ data. I don’t think it is possible to discount the possibility that MIR160a is normally expressed in the MMC, processed there and buffers the levels of ARF17. This appears to be consistent with the overall in situ data which shows that both miR160 and ARF17 are present in the majority of the ovule cells (at the same time), but miR160 reduces ARF17 mRNA levels rather than turns them off. This is also consistent with the miR160-sensitive UBI-GFP reporter that shows an overall reduction in GFP expression. Hence, Line 160/161 needs to be written carefully: “...suggesting that mature miR160 is active in a range of ovule cells, and in particular, the MMC.”

New Line 161-162: We have changed “...suggesting that the mature miR160 is active in the MMC.” into “...suggesting that mature miR160 is active in a range of ovule cells, and in particular, the MMC.”

Line 165, Fig 2m: It’s confusing how the immunolabelling of the pARF17:ARF17-GFP reporter is presented. I suggest showing the “inset image” as the main figure to allow comparisons to 2n, and show the immunolabelling as the inset.

New Line 164-167: This has been changed.

Line 167: At this point in the manuscript I still don’t think the authors have established a role for ARF17 in primary MMC “specification”. This would need to come from mutant analysis (line 185 onwards), so it seems premature to use this text here.

New Line 168: We have changed “required” to “important.”

Line 173: The authors once again refer to “proliferation of MMC”. I’m not sure whether this is a mistake or I am misunderstanding the phenotype. The majority of data points towards miR160 restricting ARF17 expression in the ovule, which prevents more than one MMC from forming. Hence, it’s a “cell specification” defect in somatic cells rather than a “cell proliferation” defect of the MMC. I suggest changing the text... “To test whether expression of ARF17 and miR160 in the MMC is required for ectopic MMCL formation we first overexpressed miRNA resistant ARF17...”

New Line 173-174: This was a mistake. We have changed “To test whether expression of *ARF17* and miR160 in the MMC promotes the proliferation of MMC, we first overexpressed *ARF17*...” into “To test whether expression of *ARF17* and miR160 is required for ectopic MMCL formation, we first overexpressed miR160-resistant *ARF17*....”

Line 182-184: Once again, the authors appear to have delivered a conclusion that is not based on the data presented. So far in the manuscript there is no evidence to suggest that “restriction of ARF17 to the MMC by miR160 is essential for MMC specification?”. The evidence from foc and pARF17:mARF17-GFP suggests that miR160 restricts ARF17 expression in the ovule and that several MMCL cells start to enlarge and accumulate more ARF17.

Line 183/4: This is an important finding because it shows that the defects observed in foc (i.e. multiple MMCL formation) are not actually dependent upon expression of ARF17 or miR160 in the MMC itself. Instead, they most likely relate to other defects in the ovule (such as the modified vasculature?).

New Line 182-184: We have changed “suggesting that restriction of *ARF17* to the MMC by miR160 is essential for MMC specification and overexpression of *ARF17* in the MMC does not promote MMC proliferation.” to “suggesting that miR160 restricts *ARF17* expression in ovule cells and overexpression of *ARF17* solely in the MMC does not promote MMC proliferation.”

Our results suggest that increasing expression of *ARF17* only in the MMC does not cause the formation of extra MMCLs. Ectopic expression of *ARF17* in other ovule cells lead them to acquiring MMC identity.

Line 186: Now we get to the experiments regarding specification!! Please correct the sentence “To test whether ARF17 is required for MMC specification....”

New Line 187: This has been changed.

Line 224: “indicated that multiple MMCs acquired MMC identity.”

New Line 224: This has been changed.

Line 228: “observed multiple MMCs accumulating callose”

New Line 229: This has been changed.

Line 219 onwards: Please define here how NPA treatments were carried out? At subsequent points in the text there are references to 1, 2 and 4 days of treatment, the significance of which is unclear.

New Line 220-221: We have changed "...thus we applied NPA to WT inflorescences for 4 days" into "...thus we continuously applied NPA to WT inflorescences every 24 hours for 4 days...."

The length of NPA treatment affects phenotypes, including number of MMCLs, position and number of auxin maxima, and the morphology of ovule. The longer the treatment, the stronger phenotypes were induced.

Line 229/230: Please consider rephrasing "Hence, disruption of PIN-dependent PAT and increased local auxin biosynthesis lead to ectopic MMC formation"

New Line 229-231: We have changed "Therefore, the PIN-dependent PAT and local biosynthesis are required for normal specification of MMC." to "Hence, disruption of PIN-dependent PAT and increased local auxin biosynthesis led to ectopic MMC formation."

Line 260-263: I don't understand this sentence. Please be precise. What is "the abnormal MMC specification"

New Line 261-263: We have changed "In summary, our findings suggest that alterations in local auxin maxima and auxin accumulation cause the abnormal MMC specification and the spatially restricted auxin activity mediated by PIN1 is required for MMC fate acquisition." to "In summary, our findings suggest that the spatially restricted auxin activity mediated by PIN1 is important for MMC fate acquisition."

Line 278 onwards: I actually think that the changes in vascular tissue (marked by PIN1 accumulation in a larger domain) may have a lot to do with the defects. They seem to correlate with wider ovules (Fig 6j,k,l) – is this the case? One concern is that the ovule in Figure 6J doesn't appear to show this broader domain of PIN1 expression in a convincing manner. The supplementary data Fig9b is much more convincing, but I am confused why the authors use such a bad ovule in 6j.

We have replaced the Fig. 6j with a new image that shows an expanded PIN1 expression domain in the central chalaza.

The expression domain of PIN1 was expanded in the central chalaza at stage 1-II in *pPIN1::PIN1-GFP foc*, *pPIN1::PIN1-GFP pARF17::mARF17*, and *pPIN1::PIN1-GFP pARF17::mARF17 foc* ovules, but we did not observe the size change of central chalaza in these ovules (Supplementary Fig. 9). At later stages, i.e., stages 2-I and 2-III, the size of central chalaza (vascular tissue) is possibly enlarged. Extra MMCPs are already produced at stage 1-II in *foc*, *pARF17::mARF17*, and *pARF17::mARF17 foc* ovules, thus altered PIN1 expression should have acted to affect MMC specification before the noticeable change of ovule size. It is still possible that changes in vascular tissue contribute to the observed defects. We are interested to test this hypothesis in the future.

Line 293-298: It's unclear to me what this part of the text is meant to be describing. Its essentially a repeat of the results from the previous sections. Perhaps I'm missing something.

New Line 294-298: Here, we described the results from NPA treatment using *spl-3*, *carf17*, and *carf17 spl1-3* mutants. These results provide evidence to support that the formation of multiple MMCLs induced by NPA depends on *ARF17* and *SPL*. We cited some results from Fig. 3 and Fig. 4 as comparisons.

Line 321: "Here we report that..."

New Line 321: This has been changed.

Line 322: The model is very difficult to interpret. The different coloured spots cannot be distinguished.

We have modified Fig. 7r by enlarging the illustration of ovule and using different symbols to represent auxin, genes, and proteins. The model is complex because many factors are involved in controlling MMC specification.

Line 324: Once again, please be cautious with the mobile miR160 conclusion (see above comments)

New Line 323-324: We have changed "the *MIR160a* gene is expressed in chalaza and funiculus, while the mature miR160 accumulates in a single hypodermal cell," into "the *MIR160a* gene is expressed in ovule cells, while the mature miR160 is particularly active in a single hypodermal cell."

Line 309, Fig 7q. Please show the qPCR data in a more appropriate form. The Figure is dominated by the mARF17 expression and it is difficult to see what happens to the other genes.

We have modified the Fig. 7q, now expression changes of all tested genes can be easily seen.

Line 353: "does not result in"

New Line 352: This has been changed.

Line 376: "a portion of them accumulate callose suggesting they are preparing to undergo meiosis"

New Line 374-375: This has been changed.

Reviewer #2 (Remarks to the Author):

I appreciate the additions made by the authors which have strengthened the manuscript. Thank

you for the efforts you have made to address my comments.

The demonstration that *foc spl* double mutants produce some MMC and FM is an important finding and corroborates the earlier reported finding that the *pARF17:mARF17* construct can partially suppress the *spl* phenotype.

More importantly for this reviewer are the elegant hybridization experiments described in supplementary figure 3. The crossing experiments are a completely orthogonal approach and clearly prove that that the *pARF17:mARF17* construct and *foc* mutation can lead to MMC production and functional megaspores (and subsequently viable female gametophytes) in an *spl* mutant background. Genetics does not lie and I find this is a nice addition. I am not aware of other *spl* suppressors and I am surprised the authors do not make this a panel of a main figure (as it is a breakthrough in my eyes) but I accept their freedom to judge for themselves.

We are so thankful for suggesting us performing these decisive tests. We totally agree with you that our new genetic results are reliable and significant in terms of supporting that *pARF17:mARF17* and the *foc* mutation can result in formation of functional MMC and FM in the *spl* null mutant background. We believe that the readers would appreciate the significance of these findings even though they are reported as a supplementary figure.

I also appreciate the addition of the *pUBI10::miR160sensor-NSL-3xGFP* data which shows that *miR160* is present in the MMC.

The requested changes to the title and text have been made and improve the clarity of the paper.

Congratulations to the authors on this nice study!

We appreciate your constructive suggestions and valuable comments that significantly strengthened our manuscripts. Thank you so very much for your support.

Reviewers' Comments:

Reviewer #1:

Remarks to the Author:

In this version of the manuscript the authors have addressed the majority of my concerns. The text has been improved to ensure consistency between sections, and several figures have been improved. The research is comprehensive and the results are very interesting - I am satisfied that this should be published.